# Activation by Interval-wise Dropout
# A Simple Way to Prevent Neural Networks from Plasticity Loss

**Sangyeon Park** [1]  **Isaac Han** [1]  **Seungwon Oh** [1]  **Kyung-Joong Kim** [1]

## Abstract

Plasticity loss, a critical challenge in neural network training, limits a model's ability to adapt to new tasks or shifts in data distribution. This paper introduces **AID** (**A**ctivation by **I**nterval-wise **D**ropout), a novel method inspired by Dropout, designed to address plasticity loss. Unlike Dropout, AID generates subnetworks by applying Dropout with different probabilities on each preactivation interval. Theoretical analysis reveals that AID regularizes the network, promoting behavior analogous to that of deep linear networks, which do not suffer from plasticity loss. We validate the effectiveness of AID in maintaining plasticity across various benchmarks, including continual learning tasks on standard image classification datasets such as CIFAR10, CIFAR100, and TinyImageNet. Furthermore, we show that AID enhances reinforcement learning performance in the Arcade Learning Environment benchmark.

## 1. Introduction

Loss of plasticity refers to the phenomenon in which a neural network loses its ability to learn, which has been reported in recent studies (Lyle et al., 2023; Dohare et al., 2021). This phenomenon has been observed in models pre-trained on datasets where labels are assigned randomly or input images are randomly permuted. When trained on a new task, these models exhibit degraded learning capabilities than freshly initialized networks. Loss of plasticity is a critical issue that arises across various domains, making it an essential problem to address. There have been evidence that plasticity loss is caused by non-stationarity (Lyle et al., 2024). While these results are from supervised domain like continual learning, plasticity loss also has been observed in

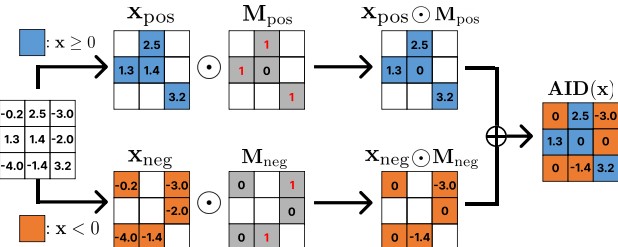

*Figure 1.* **AID Architecture for Simplified Version.** We apply Dropout at a rate of $1 - p$ for positive values and $p$ for negative values, where $m_{pos} \sim Ber(p)$ and $m_{neg} \sim Ber(1 - p)$. Unlike ReLU activation, AID allows utilizing negative preactivation and regularizing toward linear network, effectively mitigating plasticity loss.

the reinforcement learning domain, caused by its inherent non-stationarity (Sokar et al., 2023; Kumar et al., 2020). However, recent study suggests that this phenomenon is prevalent not only in non-stationary domains, but also in stationary domains (Lee et al., 2024b; Shin et al., 2024).

Several factors have been proposed as causes of plasticity loss. Earlier studies have identified issues such as dead neurons (Sokar et al., 2023) and large weight norms (Lyle et al., 2023) as potential contributors to this phenomenon. More recent research suggests that non-stationary serves as the primary trigger for plasticity loss, subsequently leading to effects like dead neurons, linearized units, and large weight norms, which constrain the network's capability and destabilize training (Lyle et al., 2024). Other factors have also been implicated, including feature rank (Kumar et al., 2020), the sharpness of the loss landscape (Lyle et al., 2023), and noise memorization (Shin et al., 2024).

Based on these insights, various approaches have been proposed to address plasticity loss. These include adding penalty terms to regularize weights (Kumar et al., 2023; Gogianu et al., 2021), resetting dead or low-utility neurons (Dohare et al., 2021; 2023; Sokar et al., 2023), shrinking the network to inject randomness (Ash & Adams, 2020; Shin et al., 2024), introducing novel architectures (Lee et al., 2024b) or novel activations(Abbas et al., 2023; Lewandowski et al., 2024), and developing new optimiza-

---

[1]Department of AI Convergence, Gwangju Institute of Science and Technology (GIST), Gwangju, South Korea. Correspondence to: Kyung-Joong Kim <kjkim@gist.ac.kr>.

*Proceedings of the 42$^{nd}$ International Conference on Machine Learning*, Vancouver, Canada. PMLR 267, 2025. Copyright 2025 by the author(s).

tion strategies (Elsayed & Mahmood, 2024).

Widely used methods such as L2 regularization and LayerNorm (Ba et al., 2016) have proven to be effective in addressing plasticity loss (Lyle et al., 2024). Several studies (Ma et al., 2023; Lee et al., 2024b) have also shown that data augmentation alleviates plasticity loss, suggesting a possible connection between overfitting and plasticity loss. However, these approaches do not tackle the underlying cause of plasticity loss. Furthermore, a recent study (Dohare et al., 2024) found that Dropout, which is commonly used to prevent overfitting, is not effective in mitigating plasticity loss. This raises the question: why is Dropout ineffective against plasticity loss? We first analyze why Dropout fails to address plasticity loss.

Building on this analysis, we propose a new method inspired by Dropout, called **AID** (**A**ctivation by **I**nterval-wise **D**ropout), to tackle plasticity loss. Unlike Dropout, AID applies interval-specific Dropout probabilities, making it functions as a nonlinear activation. Through experiments, we demonstrate that AID's masking strategy is highly effective in addressing plasticity loss. Recent studies have shown that plasticity loss does not occur in deep linear networks (Lewandowski et al., 2023; 2024; Dohare et al., 2024). We theoretically prove that AID maintains plasticity by regularizing activations to close to those of a linear network. Since AID also functions as an activation function, we theoretically prove its compatibility with He initialization, demonstrating that AID can seamlessly replace the ReLU function.

## 2. Related Works

### 2.1. Plasticity in Neural Networks

In recent years, various methods have been proposed to address plasticity loss. Several works have focused on maintaining active units (Abbas et al., 2023; Elsayed & Mahmood, 2024) or re-initializing dead units (Sokar et al., 2023; Dohare et al., 2024). Other studies have explored limiting deviations from the initial statistics of model parameters (Kumar et al., 2023; Lewandowski et al., 2023; Elsayed et al., 2024). Additionally, some methods rely on architectural modifications (Nikishin et al., 2024; Lee et al., 2024b; Lewandowski et al., 2024). Plasticity loss also occurs in the reinforcement learning due to its inherent non-stationary. Nikishin et al. (2022) proposed resetting the model, while Asadi et al. (2024) suggested resetting the optimizer state.

As noted by Berariu et al. (2021), loss of plasticity can be divided into two distinct aspects: a decreased ability of networks to minimize training loss on new data (trainability) and a decreased ability to generalize to unseen data (generalizability). While most previous works focused on trainability, Lee et al. (2024b) addressed generalizability

loss. They demonstrated that plasticity loss also occurs under a stationary distribution, as in a warm-start learning scenario where the model is pretrained on a subset of the training data and then fine-tuned on the full dataset.

Most existing studies have focused on only one of the following challenges: trainability, generalizability, or reinforcement learning. However, in this study, we validate our AID method across all three aspects, demonstrating its effectiveness in each scenario.

### 2.2. Activation Function

Our AID method is a stochastic approach similar to Dropout while also functioning as an activation function. Therefore, we aim to discuss previously proposed probabilistic activation functions. Although the field of probabilistic activation functions has not seen extensive research, two noteworthy studies exist. The first is the Randomized ReLU (RReLU) function, introduced in the Kaggle NDSB Competition (Xu, 2015). The original ReLU function maps all negative values to zero, whereas RReLU maps negative values linearly based on a random slope. During testing, negative values are mapped using the mean of the slope distribution. Their experimental results suggest that RReLU effectively prevents overfitting. Another example of a probabilistic activation function is DropReLU (Liang et al., 2021). DropReLU randomly determines whether a node's activation is processed through a ReLU function or a linear function. The authors claim that DropReLU improves the generalization performance of neural networks. The fundamental distinction between these probabilistic activation functions and our method lies in the generality of our approach. Unlike simple probabilistic activation functions, our method encompasses techniques such as Dropout and ReLU, providing a more comprehensive framework.

Another related approach involves activation functions designed to address plasticity loss. (Abbas et al., 2023) proposed the Concatenated Rectified Linear Units (CReLU), which concatenates the outputs of the standard ReLU applied to the input and its negation. This structure prevents the occurrence of dead units, thereby improving plasticity. Additionally, trainable activation functions have also been shown to effectively mitigate plasticity loss in reinforcement learning (Delfosse et al., 2024). Specifically, they introduced a trainable rational activation function that prevents value overfitting and overestimation in reinforcement learning.

## 3. Why is Dropout Ineffective for Maintaining Plasticity?

In this section, we investigate the ineffectiveness of Dropout in mitigating plasticity loss. Dropout operates by randomly

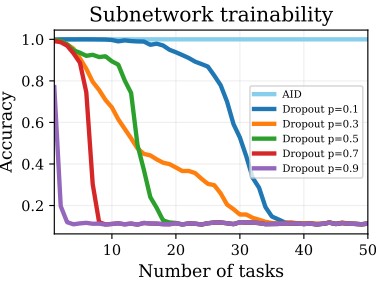 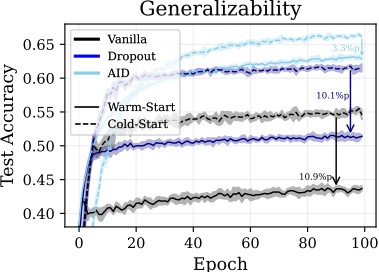

*Figure 2.* **Left.** Random label MNIST experiment using an 8-layer MLP. Higher dropout probabilities result in significant trainability loss. **Middle.** Accuracy of the subnetworks trained on random target. Each subnetworks are sampled from original network after each epoch. Subnetworks of the Dropout also experience trainability loss. **Right.** Warm-start scenario of Resnet-18 model with CIFAR100 dataset. Dropout improves generalization performance; however, the reduction in accuracy compared to the cold-start scenario is nearly identical to that of the vanilla model.

deactivating specific nodes during training, effectively generating random subnetworks. This mechanism fosters an ensemble effect, where the network learns more generalized features across these subnetworks. However, our hypothesis suggests that this process does not address plasticity loss:

*"Dropout merely creates multiple subnetworks, all of which suffer from plasticity loss."*

In other words, because each subnetwork independently experiences plasticity loss, the combined network—treated as an ensemble—also suffers from degraded plasticity.

To validate this hypothesis, we conducted experiments on two forms of plasticity loss: trainability and generalizability. Details of the experimental configurations are provided in Appendix F.1.

### 3.1. Trainability Perspective

To examine the relationship between Dropout and trainability, we trained an 8-layer MLP on the randomly labeled MNIST dataset. For each task, labels were shuffled, and the network was trained for 100 epochs. Following prior studies (Lyle et al., 2023; Kumar et al., 2023), trainability degradation was quantified as the accuracy drop relative to the first task. Figure 2 (left) demonstrates that higher Dropout probabilities lead to more pronounced trainability degradation. This observation aligns with prior findings that Dropout exacerbates plasticity loss (Dohare et al., 2024). As Dropout probabilities increase, the network is divided into progressively smaller subnetworks, reducing the expected number of parameters available for learning. This is consistent with the results of Lyle et al. (2023), which show that smaller networks are more vulnerable to plasticity loss. To further explore this phenomenon, we measured the trainability of individual subnetworks. At the end of each task, we sampled 10 subnetworks and trained them on new tasks. As shown in Figure 2 (middle), subnetworks with higher Dropout probabilities exhibited more severe train-

ability degradation, reinforcing our hypothesis that Dropout creates multiple subnetworks, each of which suffers from plasticity loss. In contrast, our proposed method, AID, effectively maintained trainability across all tasks, demonstrating its robustness against plasticity loss.

### 3.2. Generalizability Perspective

To evaluate generalizability, we conducted a warm-start learning experiment inspired by Lee et al. (2024b). Specifically, we pre-trained a RESNET-18 model on 10% of the training data for 1,000 epochs before continuing training on the full dataset. We repeated this experiment for models trained with vanilla settings, Dropout, and AID. The results are shown in Figure 2 (right). Dropout appeared to improve overall performance, with both warm-start and cold-start models showing better accuracy compared to the vanilla setting. However, we argue that this improvement arises from enhanced model generalization rather than improved generalizability. Dropout fails to reduce the accuracy gap between warm-start and cold-start training (10.1%p) as vanilla setting does (10.9%p). If Dropout were primarily addressing plasticity loss, we would expect disproportionately higher performance improvements in warm-start models, as cold-start models inherently maintain perfect plasticity. In contrast, the performance gap with AID was significantly smaller (3.3%p compared to the vanilla model's 10.9%p). This observation suggests that AID effectively mitigates plasticity loss, as warm-start models trained with AID retain a higher degree of plasticity compared to those trained with Dropout. Extended results regarding generalizability can be found in Appendix G.2.1.

## 4. Method

### 4.1. Notations

We denote $r(\cdot)$ to be a ReLU (Rectified Linear Unit) activation function, which means that $r(x) = \max(x, 0)$ for input

data $x$. For ease of notation, we define a negative ReLU, $\bar{r}(x) = \min(x, 0)$. Furthermore, for a real number $\alpha > 0$, we define modified leaky ReLU, $r_\alpha$ as a function that scales the positive part of the input by $\alpha$, and the negative part by $1 - \alpha$, formally given as $r_\alpha(x) = \frac{1}{2}x + (\alpha - \frac{1}{2})|x|$ (e.g. $r_1(x) = r(x)$, $r_0(x) = \bar{r}(x)$, and $r_{1/2}(x) = \frac{1}{2}x$). Note that, $r_\alpha$ differs from leaky ReLU, since its slope on the positive side may not be 1.

## 4.2. Activation by Interval-wise Dropout (AID)

Activation by Interval-wise Dropout (AID) applies different Dropout probabilities to distinct intervals of preactivation values, functioning as a non-linear activation. Specifically, given $k$ predefined intervals and their corresponding Dropout probabilities $\{p_1, p_2, \ldots, p_k\}$, AID generates $k$ distinct Dropout masks. Each mask is applied only to the values falling within its corresponding interval, while other intervals are unaffected by that mask. Formally, let the predefined intervals be $\{I_1, I_2, \ldots, I_k\}$, where $I_j = [l_j, u_j)$ for $j = 1, 2, \ldots, k$. The union of these intervals covers the entire range of real values, i.e., $\bigcup_{j=1}^{k} I_j = \mathbb{R}$. Moreover, the intervals are disjoint, meaning there is no overlap between any two intervals, i.e., $I_i \cap I_j = \emptyset$ for all $i \neq j$. AID operates as a non-linear activation function when the Dropout probabilities vary across intervals, i.e., when $p_i \neq p_j$ for some $i \neq j$. If all Dropout probabilities are identical ($p_1 = p_2 = \cdots = p_k$), AID reduces to a standard Dropout mechanism. The output of the AID mechanism for an input vector $\mathbf{x}$ is given by: $AID(\mathbf{x}) = \mathbf{x} \odot \mathbf{m}$. Where $\mathbf{m}$ is the Dropout mask vector defined as: $m_i = 0$ with probability $p_j$ if $x_i \in I_j$, and $m_i = 1$ otherwise. This interval-wise Dropout mechanism enables the model to handle varying ranges of preactivation values, effectively functioning as a nonlinear activation function while applying stochastic regularization with different probabilities across intervals.

During the testing phase of traditional Dropout, each weight is scaled by the probability $1 - p$, which is the chance that a node is not dropped. Similarly, in AID, as each interval has a unique Dropout probability, the preactivation values within each interval $I_j$ are scaled by $1 - p_j$ for testing. The pseudo-code for AID is presented in Algorithm 1.

AID generalizes concepts from ReLU, standard Dropout, and DropReLU. We provide a detailed discussion of these relationships in Appendix C.1. Since defining each interval and its corresponding Dropout probability requires exploring a vast hyperparameter space, Algorithm 1 demands extensive hyperparameter tuning. To address this, we introduce a simplified version of AID in the next section.

## 4.3. Simplified Version of AID

Section 4.2 shows that this methodology offers infinitely many possible configurations depending on the number and

---

**Algorithm 1** Activation by Interval-wise Dropout (AID)

**Given:** Predefined intervals $\{I_1, I_2, \ldots, I_k\}$, Dropout probabilities $\{p_1, p_2, \ldots, p_k\}$
**Input:** Preactivation values $\mathbf{x} \in \mathbb{R}^n$
**Output:** Postactivation values $\mathbf{y} \in \mathbb{R}^n$
**if** training phase **then**
    Initialize $\mathbf{y} \in \mathbb{R}^n$ as an empty vector
    **for** each $x_i \in \mathbf{x}$ **do**
        Identify interval $I_j$ such that $x_i \in I_j$
        Initialize binary variable $m \sim Bernoulli(1 - p_j)$
        Apply mask: $y_i = x_i \cdot m$
    **end for**
**else if** test phase **then**
    Initialize $\mathbf{y} \in \mathbb{R}^n$ as an empty vector
    **for** each $x_i \in \mathbf{x}$ **do**
        Identify interval $I_j$ such that $x_i \in I_j$
        Apply scaling: $y_i = x_i \cdot (1 - p_j)$
    **end for**
**end if**
**return** $\mathbf{y}$

---

range of intervals, as well as the Dropout probabilities. To reduce the hyperparameter search space and improve computational efficiency, we propose the Simplified AID. Specifically, we define $AID_p(\cdot)$ as a function that applies a Dropout rate of $1 - p$ to positive and $p$ to negative values. This definition is intuitive since $AID_p$ behaves like ReLU when $p = 1$ and a linear network when $p = 0.5$. Additionally, we will demonstrate that this simplified version has an effect that regularizes to linear network and an advantage when He initialization is applied, as discussed in Section 4.4. Next, we explore a property of $AID_p$ that are useful for its practical implementation.

*Property* 1. Applying $AID_p(\cdot)$ is equivalent to applying $r(\cdot)$ with probability $p$, and $\bar{r}(\cdot)$ with probability $1 - p$.

We provide a brief proof in Appendix A. Using Property 1, the implementation of AID becomes straightforward. An important consideration is whether scaling up is performed during training. Unlike the inverted implementation of Dropout, which behaves as the identity function during the test phase, AID is intended to operate as an activation function and thus should retain its nonlinearity at test time at test time. To ensure this, we adopt the standard implementation, following the approach of Srivastava et al. (2014). Specifically, AID utilizes $r_\alpha(\cdot)|_{\alpha=p}$, which is the same as passing the average values from the training phase. To clarify the process, we provide the pseudo-code for the AID activation in Algorithm 2.

Algorithm 2 demonstrates that AID can be easily applied by replacing the commonly used ReLU activation. Moreover, it shares the same computational complexity as standard Dropout. These properties enable AID to integrate

**Algorithm 2** Simplified Version of AID

---

**Given:** ReLU $r$, negative ReLU $\bar{r}$, modified leaky ReLU $r_\alpha$ and one vector $\mathbf{1} \in \mathbb{R}^n$
**Input:** Preactivation values $\mathbf{x} \in \mathbb{R}^n$, coefficient $p$
**Output:** Postactivation values $\mathbf{y} \in \mathbb{R}^n$
**if** training phase **then**
    $\mathbf{M} \leftarrow$ Generate binary masks from $Bernoulli(p)$
    $\mathbf{y} \leftarrow \mathbf{M} \odot r(\mathbf{x}) + (\mathbf{1} - \mathbf{M}) \odot \bar{r}(\mathbf{x})$
**else if** test phase **then**
    $\mathbf{y} \leftarrow r_p(\mathbf{x})$
**end if**
**return y**

---

seamlessly with various algorithms and model architectures, regardless of their structure. For reproducibility, we provide the PyTorch implementation of the AID module in Appendix H. Throughout this paper, AID is implemented as described in Algorithm 2, unless specified otherwise. In the next section, we study the factors that contribute to the effectiveness of AID.

### 4.4. Theoretical Analysis

Recent studies (Dohare et al., 2024; Lewandowski et al., 2024) have found that linear networks do not suffer from plasticity loss and provided theoretical insights into this phenomenon. This highlights that the properties of linear networks play a crucial role in resolving the issue of plasticity loss.

From an intuitive perspective, AID shares properties with deep linear networks in that, it provides the possibility of linearly passing all preactivation values and having gradients of 1. This behavior contrasts with activations such as ReLU, RReLU, and leaky ReLU, which tend to pass fewer or no values for negative preactivation. Building on this intuition, the following theoretical analysis establishes that AID as a regularizer, effectively constraining the network to exhibit properties of a linear network.

**Theorem 4.1.** *For a 2-layer network, AID has a regularization effect that constrains the model to behave like a linear network. Formally, let weight matrices be $\mathbf{W}_1, \mathbf{W}_2 \in \mathbb{R}^{n \times n}$, the input and target vectors be $\mathbf{x}, \mathbf{y} \in \mathbb{R}^n$, and the coefficient of AID be $p$. Then, for learnable parameter $\theta$:*

$$L_p^{AID}(\theta) \doteq \mathbb{E}\big[\|\mathbf{W}_2 AID_p(\mathbf{W}_1\mathbf{x}) - \mathbf{y}\|_2^2\big] \quad (1)$$

$$L_p(\theta) \doteq \|\mathbf{W}_2 r_p(\mathbf{W}_1\mathbf{x}) - \mathbf{y}\|_2^2$$

$$R_p(\theta) \doteq \|\mathbf{W}_2\big(\frac{1}{2}\mathbf{W}_1\mathbf{x}\big) - \mathbf{W}_2 r_p(\mathbf{W}_1\mathbf{x})\|_2^2$$

$$\implies L_p^{AID}(\theta) \geq L_p(\theta) + \frac{4p(1-p)}{n(2p-1)^2}R_p(\theta).$$

We provide the detailed proof in Appendix B, following

the approach taken by Liang et al. (2021). Theorem 4.1 shows that the lower bound of Equation (1)—the loss under AID—can be decomposed into two terms: the loss under a modified leaky ReLU ($L_p$), and a regularization term ($R_p$) that encourages linearity in the network. As the hyperparameter $p$ approaches 0.5, AID imposes stronger regularization toward linear behavior. Prior work (Lewandowski et al., 2024) has shown that linear networks do not suffer from trainability issues, suggesting that the linearity-inducing effect of AID directly contributes to mitigating plasticity loss. On the other hand, simply adding a Dropout layer with ReLU does not yield the same effect, and the corresponding lemma is provided in Appendix C.2.

Since AID functions as an activation, it is important to examine its compatibility with existing network initialization methods designed for activations. Therefore, we show that AID ensures the same stability as the ReLU function when used with Kaiming He initialization. The following property illustrates this strength:

*Property* 2. Replacing ReLU activation with $AID_p$ ensures the same stability during the early stages of training when using Kaiming He initialization.

Property 2 guarantees that the output and gradient will not explode or vanish, thereby supporting the learning stability of the network during its initial training. The outline of the explanation builds on He et al. (2015)'s formulation, utilizing the fact that replacing ReLU with AID does not alter the progression of the equations. A detailed explanation is provided in Appendix D.

## 5. Experiments

In this section, we empirically evaluate the effectiveness of the proposed method across various tasks. First, we compare AID with previous methods in a range of continual learning tasks to assess its impact on mitigating plasticity loss in Section 5.1. Second, we investigate whether AID provides tangible benefits in reinforcement learning in Section 5.2. Finally, we demonstrate that AID improves generalization in standard classification tasks in Section 5.3.

### 5.1. Continual Learning

We compare AID with various methods that were proposed to address the challenges of maintaining plasticity. In addition, Randomized ReLU and DropReLU are included due to their structural similarity to our method. Refer to Appendix E for the description of each baseline.

- **Regularization**: Dropout (Srivastava et al., 2014), L2 regularization (L2) (Krogh & Hertz, 1991) and L2 regularization to initial value (L2 Init) (Kumar et al., 2023)

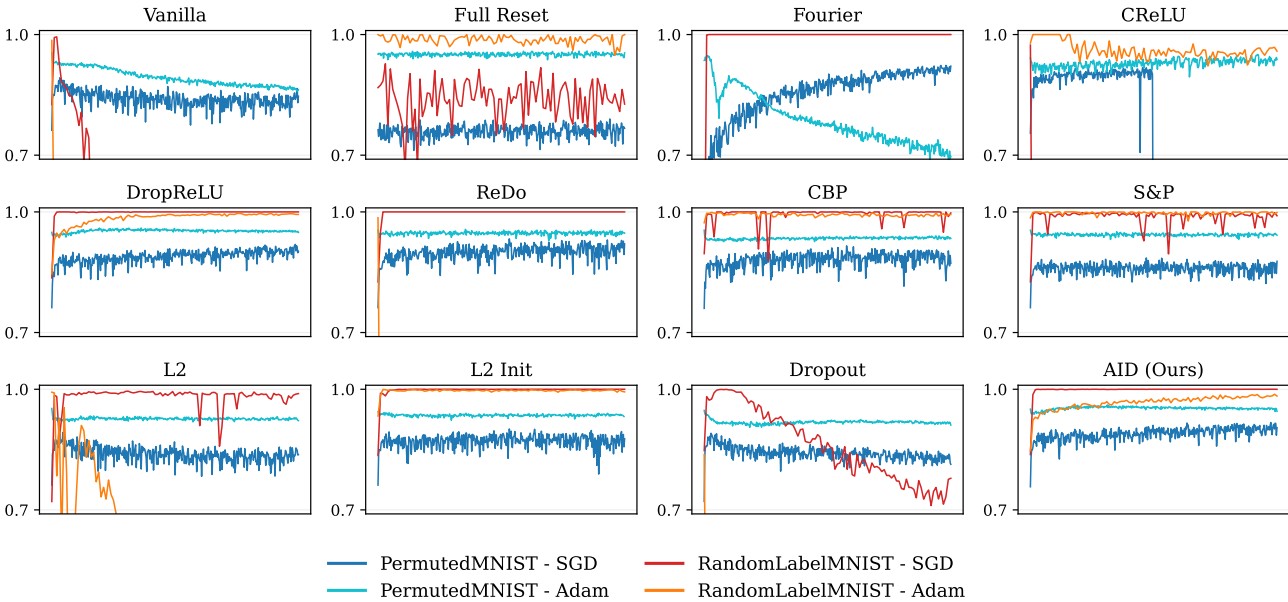

*Figure 3.* **Results for Trainability Experiments.** We plot the train accuracy comparisons across different optimizers on Permuted MNIST and Random Label MNIST for each method, with the x-axis representing tasks. For visibility, we did not show the region using standard deviation. Notably, AID consistently achieves high accuracy across all conditions, demonstrating its robustness in maintaining trainability.

- **Re-initialization**: Shrink & Perturb (S&P) (Ash & Adams, 2020), Recycling Dormant neurons (ReDo) (Sokar et al., 2023) and Continual Backprop (CBP) (Dohare et al., 2024)

- **Activation function**: Concatenated ReLU (CReLU) (Abbas et al., 2023), Randomized ReLU (RReLU) (Xu, 2015), deep Fourier features (Fourier) (Lewandowski et al., 2024), and DropReLU (Liang et al., 2021)

### 5.1.1. TRAINABILITY

We start by validating the AID's ability to maintain trainability. Lee et al. (2024a) categorized the trainability of neural networks into two cases: Input trainability and Label trainability[1]. Input trainability denotes the adaptability to changing of input data, and label trainability denotes the adaptability to evolving input-output relationships. Under the concept, we conducted permuted MNIST experiment for input plasticity by randomly permuting the input data and random label MNIST experiment for label plasticity by shuffling the labels at each task. Detailed experimental settings are shown in Appendix F.2.

Figure 3 shows the training accuracy for each method, categorized by optimizer and experimental settings. We used learning rates of $3\mathrm{e}{-2}$ for SGD and $1\mathrm{e}{-3}$ for Adam optimizer. The results reveal several key findings. First, training

accuracy of vanilla model gradually declines for both SGD and Adam optimizers. In contrast, methods specifically designed to address plasticity loss—such as Fourier, CReLU, ReDo, and CBP—demonstrate improved plasticity under most conditions. However, these methods fail to retain accuracy in certain conditions. In particular, S&P and L2 Init often struggle to maintain plasticity when trained with a low learning rate (see Appendix G.1.1), suggesting that their effectiveness may be sensitive to learning rate choices. Notably, Dropout fails to mitigate plasticity loss in most cases compared to the vanilla model, supporting the results of Dohare et al. (2024). In contrast, DropReLU and AID consistently achieve high accuracy across all settings, which can be attributed to the regularizing effect of the linear network, as discussed in Section 4.4. We provide detailed experimental results with various learning rates and methods, as well as additional metrics relevant to plasticity loss—including dormant neuron rate, effective rank, and average unit sign entropy—for Vanilla, Dropout, and AID in Appendix G.1.

### 5.1.2. GENERALIZABILITY

In our earlier experiments, we confirmed that AID effectively maintains ability to adapt new data. However, while adaptability is important, it is even more critical to ensure that the model retain the ability to generalize well to unseen data. Therefore, generalizability is a critical aspect related to plasticity loss. We evaluated the generalizability of AID across three benchmark settings: continual-full, continual-limited, and class-incremental. For this evalu-

---

[1]In Lee et al. (2024a), the terms input plasticity and label plasticity were originally used. For consistency with our study, we instead adopt the terms input trainability and label trainability.

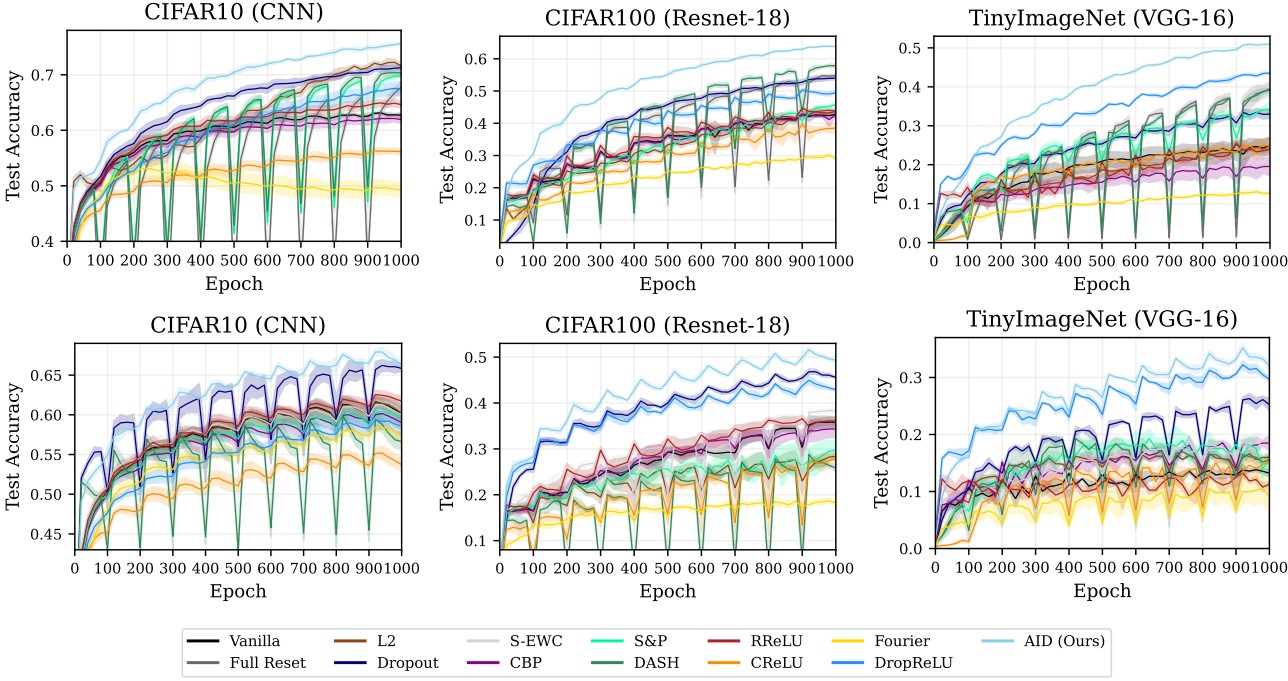

*Figure 4.* **Results on Continual Full & Limited Settings.** The dataset was divided into 10 chunks, with models trained over 10 stages of 100 epochs each. The settings are distinguished by whether access to previously seen data is allowed (**top row, continual full**) or not (**bottom row, continual limited**). AID consistently outperforms baseline methods, highlighting its effectiveness in mitigating plasticity loss across both settings.

ation, we utilized three datasets: CIFAR10, CIFAR100 (Krizhevsky et al., 2009), and TinyImageNet (Le & Yang, 2015).

**Continual Full & Limited**  We investigated two continual learning settings inspired by previous works (Lee et al., 2024b; Shen et al., 2024). In the continual full setting, an extended version of the warm-start framework, training progress is divided into 10 phases, with 10% of the dataset introduced at each phase while maintaining access to all previously seen data throughout training. In contrast, the continual limited setting imposes stricter constraints by restricting access to past data, reflecting practical challenges such as privacy concerns and storage limitations. In our experiments, we exclude L2 Init and ReDo, as they have been shown to have no improvement on generalizability, and instead include the recently proposed Direction-Aware SHrinking (DASH) (Shin et al., 2024) as a baseline. Additionally, we incorporate Streaming Elastic Weight Consolidation (S-EWC) (Kirkpatrick et al., 2017; Elsayed & Mahmood, 2024) on continual limited, which has demonstrated effectiveness in mitigating forgetting under constrained settings.

The results of the Continual setting are presented in Figure 4. Several methods—such as DASH and S&P—achieve comparable or even superior performance to full reset, de-

pending on the specific task and dataset. However, AID demonstrates a noticeable performance advantage over other methods throughout the entire training process. This result highlights that plasticity can be maintained without the need to re-initialize certain layers or neurons, which carries the risk of losing previously learned knowledge. Notably, this trend persists even in the continual limited setting. Interestingly, methods belonging to the Dropout family (Dropout, DropReLU, and AID) also exhibit strong performance in this setting, aligning with previous findings that suggest Dropout can help mitigate catastrophic forgetting (Mirzadeh et al., 2020). We evaluated the sensitivity of AID to its hyperparameter in the continual full setting with results confirming its robustness provided in Appendix G.2.2.

**Class-Incremental**  In addition to these continual learning settings, we also conducted experiments on class-incremental Learning, inspired by previous works (Lewandowski et al., 2024; Dohare et al., 2024). Unlike the warm-start framework, this approach partitions the training process into 20 phases based on class labels, incrementally introducing new classes at each phase. Like continual full setting, the model is trained on all available data, including previously introduced class data. Further details regarding the experimental setup can be found in Appendix F.3.

Figure 5 presents the difference in test accuracy between

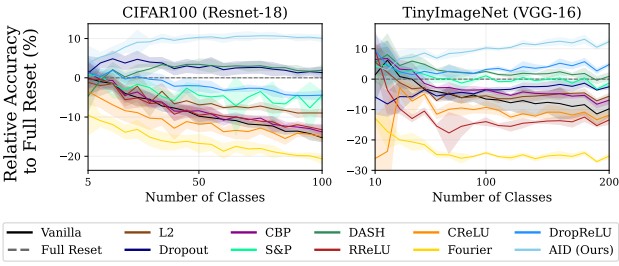

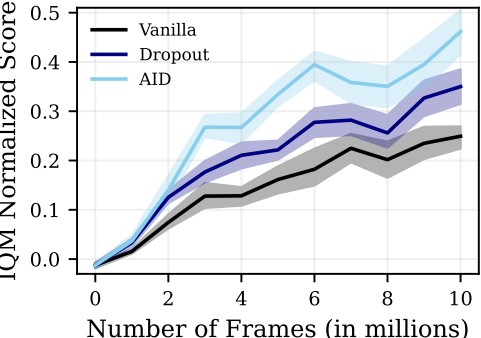

*Figure 5.* **Results on Class-Incremental Setting.** The figure shows the relative accuracy compared to a model trained from scratch. AID consistently outperforms the full reset approach, maintaining its advantage throughout the entire training process.

a model trained with full reset as each class is introduced. Methods—such as Dropout and DASH—initially demonstrate an accuracy advantage when full reset is applied during training. However, as training progresses, this advantage diminishes, highlighting their limitations in long-term plasticity retention. In contrast, AID stands out as the only method that consistently maintains its advantage over full reset throughout the training process. These results suggest that AID effectively mitigates plasticity loss across various experiments evaluating generalizability, reinforcing its robustness in continual learning scenarios. Additional figures showing overall test accuracy are provided in Appendix G.3.

### 5.2. Reinforcement Learning

A high replay ratio is known to offer high sample efficiency in reinforcement learning. However, increasing the replay ratio often leads to overfitting to early data and results in plasticity loss, as highlighted in previous studies (Kumar et al., 2020; Nikishin et al., 2022). In this section, we demonstrate that using AID can effectively mitigate these challenges by maintaining plasticity while benefiting from high sample efficiency.

The experimental setup is as follows: Unlike standard DQN (Mnih et al., 2015) use $RR = 0.25$, we train with $RR = 1$ on several Atari (Bellemare et al., 2013) tasks, using the CleanRL environment (Huang et al., 2022). We use the 17 games to train and hyperparameter settings from Sokar et al. (2023). Due to the high computational cost associated with a high RR, we followed the settings of Sokar et al. (2023); Elsayed et al. (2024), where the models are trained for 10 millions frames.

The IQM Human Normalized Score results for the 17 Atari games are presented in Figure 6, with shading indicating stratified bootstrap 95% confidence intervals. The results demonstrate that AID significantly improves sample efficiency compared to the vanilla model. Traditionally, Dropout has been rarely used in reinforcement learning due

*Figure 6.* **Results on Reinforcement Learning.** We train DQN model with a replay ratio of 1. The plot presents the IQM Human Normalized Score for 17 Atari games, with shaded regions representing the 95% confidence intervals across 5 random seeds. The results indicate that AID significantly improves sample efficiency compared to the vanilla model.

to several factors (Hausknecht & Wagener, 2022). Additionally, as demonstrated in Section 5.1.1, its limitation in mitigating plasticity loss in non-stationary environments may further explain this issue. We note that the slight performance improvement observed with a small Dropout rate appears to result from the ensemble effect of Dropout, rather than from mitigation of plasticity loss. In contrast, AID achieves performance gains by effectively addressing plasticity loss itself. Further discussion and additional results on this point are provided in Appendix G.4.

### 5.3. Standard Supervised Learning

The preceding experiments have demonstrated that AID effectively addresses the loss of plasticity and exhibits its effectiveness in non-stationary problems. In this section, we investigate whether AID also provides advantages in classical deep learning scenarios without non-stationarity.

For comparison, we utilized L2 regularization and Dropout, which are commonly employed in supervised learning, as baselines. The model was trained for a total of 200 epochs using the Adam optimizer with a learning rate of 0.001. The learning rate was reduced by a factor of 10 at the 100th and 150th epochs. The final test accuracy is shown in Table 1.

*Table 1.* **Results on Generalization with Learning Rate Decay.** The final test accuracy after training 200 epochs. The learning rate initialized with 0.001 and divided by 10 at epoch 100 and 150.

| Method | CIFAR10 (CNN) | CIFAR100 (Resnet-18) | TinyImageNet (VGG-16) |
|---|---|---|---|
| Vanilla | $0.660 \pm 0.0058$ | $0.581 \pm 0.0029$ | $0.430 \pm 0.0009$ |
| L2 | $0.742 \pm 0.0038$ | $0.574 \pm 0.0017$ | $0.417 \pm 0.0042$ |
| Dropout | $0.733 \pm 0.0037$ | $0.651 \pm 0.0028$ | $0.466 \pm 0.0040$ |
| AID | $\mathbf{0.748 \pm 0.0021}$ | $\mathbf{0.690 \pm 0.0014}$ | $\mathbf{0.519 \pm 0.0043}$ |

The results presented in Table 1 demonstrate that AID is effective not only in addressing plasticity loss but also in improving generalization performance in classical supervised learning settings. Interestingly, AID reduces the generalization gap more effectively than standard methods commonly used to addressing overfitting. We report the corresponding learning curves in Appendix G.5.

## 6. Conclusion

In this paper, we analyzed why Dropout fails to maintain plasticity. From this observation, we proposed AID (Activation by Interval-wise Dropout) and established through theoretical analysis that it acts as a regularizer by penalizing the difference between nonlinear and linear networks. Through extensive experiments, we demonstrated the effectiveness of AID across various continual learning tasks, showing its superiority in maintaining plasticity and enabling better performance over existing methods. Moreover, we validated its versatility by showing that AID performs well even in classical deep learning tasks, highlighting its generalizability across diverse scenarios.

Despite its effectiveness, several limitations remain in the current implementation of AID. Our experiments primarily focused on configurations with a simplified version, leaving the potential impact of more complex configurations underexplored. While we have provided a theoretical explanation for the high trainability of AID, its relationship to the observed improvements in generalizability remains uncertain. Future work will address this limitation by optimizing AID's scalability, exploring diverse configurations and adapting it to new architectures. Additionally, investigating the connection between trainability and generalizability would be a promising direction for future research.

## Acknowledgements

This work was supported by Institute of Information & communications Technology Planning & Evaluation (IITP) grant funded by the Korea government (MSIT) (No.2019-0-01842, Artificial Intelligence Graduate School Program (GIST)).

This work was supported by the GIST-MIT Research Collaboration grant funded by the GIST in 2025.

We appreciate the high-performance GPU computing support of HPC-AI Open Infrastructure via GIST SCENT.

This research was financially supported by the Ministry of Trade, Industry and Energy(MOTIE) and Korea Institute for Advancement of Technology(KIAT) through the "International Cooperative R&D program" (Grant No. P0028435)

## Impact Statement

The proposed method, Activation by Interval-wise Dropout (AID), improves neural network adaptability by mitigating plasticity loss—a critical challenge in continual and reinforcement learning. This enhancement enables better generalization in dynamic environments, making AID applicable across both domains. These advancements contribute to more robust AI systems capable of efficiently handling real-world tasks, such as autonomous systems and personalized user experiences.

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

## A. Proof of Property 1

*Proof.* Consider an input vector $\mathbf{x} = (x_1, x_2, \ldots, x_n) \in \mathbb{R}^n$. After applying $AID_p$ to $\mathbf{x}$, the $i$-th element is expressed as:

$$AID_p(\mathbf{x})_i = \begin{cases} p_i x_i & x_i \geq 0, \\ (1 - p_i) x_i & x_i < 0, \end{cases}$$

where $p_i$ is a random variable sampled from $Ber(p)$.

To enable vectorized computation, we introduce a mask matrix $\mathbf{M} \in \mathbb{R}^{n \times n}$, satisfying $AID_p(\mathbf{x}) = \mathbf{M}\mathbf{x}$. Specifically, the diagonal elements $m_{(i,i)}$ of $\mathbf{M}$ are sampled as follows:

$$m_{(i,i)} \sim Ber((2p - 1)H(x_i) + (1 - p)),$$

where $H(\cdot)$ is the step function, mapping positive values to 1 and negative values to 0. Then, the matrix $\mathbf{M}$ can be written as:

$$\mathbf{M} = (2\mathbf{P} - \mathbf{I})H(\mathbf{x}) + (\mathbf{I} - \mathbf{P}) \tag{2}$$

where $\mathbf{I}$ is the identity matrix of size $n \times n$ and $\mathbf{P} \doteq \text{diag}((p_1, p_2, \ldots, p_n))$, where $\text{diag}(\cdot)$ is the function that creates a matrix whose diagonal elements are given by the corresponding vector.

Using equation 2, the AID operation can be expanded as:

$$\begin{aligned} AID_p(\mathbf{x}) &= \mathbf{M}\mathbf{x} \\ &= [(2\mathbf{P} - \mathbf{I})H(\mathbf{x}) + (\mathbf{I} - \mathbf{P})]\mathbf{x} \\ &= (2\mathbf{P} - \mathbf{I})r(\mathbf{x}) + (\mathbf{I} - \mathbf{P})\mathbf{x} \\ &= (2\mathbf{P} - \mathbf{I})r(\mathbf{x}) + (\mathbf{I} - \mathbf{P})(\mathbf{x} - r(\mathbf{x}) + r(\mathbf{x})) \\ &= \mathbf{P}r(\mathbf{x}) + (\mathbf{I} - \mathbf{P})(\mathbf{x} - r(\mathbf{x})) \\ &= \mathbf{P}r(\mathbf{x}) + (\mathbf{I} - \mathbf{P})\bar{r}(\mathbf{x}) \\ &= [\mathbf{P}r + (\mathbf{I} - \mathbf{P})\bar{r}]\mathbf{x}. \end{aligned} \tag{3}$$

Thus, applying AID is equivalent to applying ReLU with probability $p$, and negative ReLU with probability $1 - p$. □

# B. Proof of Theorem 4.1

*Proof.* We clarify that we follow a similar procedure to the one demonstrated during the proof by Liang et al. (2021), which employs a comparable methodology.

Let the $m$ input data points and true labels be $\mathbf{x}_1, \mathbf{x}_2, \ldots, \mathbf{x}_m$ and $\mathbf{y}_1, \mathbf{y}_2, \ldots, \mathbf{y}_m$ ($\mathbf{x}_i, \mathbf{y}_i \in \mathbb{R}^n$). For simplicity, let weight matrices be $\mathbf{W}_1, \mathbf{W}_2 \in \mathbb{R}^{n \times n}$ and they do not have bias vectors. If we assume that this model uses $AID_p$, using notation in Equation (3), the prediction of the model $\hat{\mathbf{y}}_i$ can be expressed as below:

$$\hat{\mathbf{y}}_i = \mathbf{W}_2 \left( \mathbf{I} - \mathbf{P} + (2\mathbf{P} - \mathbf{I})r \right) (\mathbf{W}_1 \mathbf{x}_i)$$

For testing, using $\mathbb{E}[\mathbf{P}] = p\mathbf{I}$, we get $\hat{\mathbf{y}}_i = \mathbf{W}_2((1-p)\mathbf{I} + (2p-1)r)(\mathbf{W}_1 \mathbf{x}_i)$. Here, $\mathbf{I}$, $\mathbf{P}$, $r(\cdot)$ and $r_p(\cdot)$ refer to the identity matrix, Bernoulli mask, ReLU and modified leaky ReLU function, respectively, as mentioned in Property 1.

Then, the training process can be formally described as below:

$$\min_{\mathbf{W}_1, \mathbf{W}_2} \sum_{i=1}^{m} \mathbb{E} \left[ \| \mathbf{W}_2 \left( \mathbf{I} - \mathbf{P} + (2\mathbf{P} - \mathbf{I})r \right) (\mathbf{W}_1 \mathbf{x}_i) - \mathbf{y}_i \|_2^2 \right]$$

For simplicity, the input vector and label will be denoted as $(\mathbf{x}, \mathbf{y})$ in the following proof. Let $\mathbf{D} \doteq diag((\mathbf{W}_1 \mathbf{x} > 0))$ be the diagonal matrix, where $(\mathbf{W}_1 \mathbf{x} > 0)$ is 0-1 mask vector, which means $r(\mathbf{W}_1 \mathbf{x}) = \mathbf{D} \mathbf{W}_1 \mathbf{x}$. To reduce the complexity of the equations, let us denote $\mathbf{S}$, $\mathbf{S}_p$ and $\mathbf{v}$ as below:

$$\mathbf{S} \doteq \mathbf{I} - \mathbf{P} + (2\mathbf{P} - \mathbf{I})\mathbf{D}$$
$$\mathbf{S}_p \doteq (1-p)\mathbf{I} + (2p-1)\mathbf{D}$$
$$\mathbf{v} \doteq \mathbf{W}_1 \mathbf{x}$$

Note that $S$ and $S_p$ are diagonal matrix. Let the tr be trace of matrix and $\text{vec}(\cdot)$ be the function that makes elements of diagonal matrix to the vector. Then, we can expand the objective function as below:

$$
\begin{aligned}
\mathbb{E}\left[ \| \mathbf{W}_2 \left( \mathbf{I} - \mathbf{P} + (2\mathbf{P} - \mathbf{I})r \right) (\mathbf{W}_1 \mathbf{x}) - \mathbf{y} \|_2^2 \right] &= \mathbb{E}[\| \mathbf{W}_2 \mathbf{S} \mathbf{v} - \mathbf{y} \|_2^2] \\
&= \mathbb{E}\left[ \text{tr}(\mathbf{W}_2 \mathbf{S} \mathbf{v} \mathbf{v}^\top \mathbf{S} \mathbf{W}_2^\top) \right] - 2\text{tr}(\mathbf{W}_2 \mathbf{S}_p \mathbf{v} \mathbf{y}^\top) + \text{tr}(\mathbf{y} \mathbf{y}^\top)
\end{aligned}
\tag{4}
$$

where, $\mathbb{E}\left[ \text{tr}(\mathbf{W}_2 \mathbf{S} \mathbf{v} \mathbf{v}^\top \mathbf{S} \mathbf{W}_2^\top) \right]$

$$
\begin{aligned}
&= \mathbb{E}\left[ \text{tr}(\text{vec}(\mathbf{S})\text{vec}(\mathbf{S})^\top \text{diag}(\mathbf{v}) \mathbf{W}_2^\top \mathbf{W}_2 \text{diag}(\mathbf{v})) \right] \\
&= \text{tr}(\mathbb{E}[\text{vec}(\mathbf{S})\text{vec}(\mathbf{S})^\top] \text{diag}(\mathbf{v}) \mathbf{W}_2^\top \mathbf{W}_2 \text{diag}(\mathbf{v})) \\
&= \text{tr}(\mathbb{E}[\text{vec}(\mathbf{S}_p)\text{vec}(\mathbf{S}_p)^\top + \text{vec}(\mathbf{S})\text{vec}(\mathbf{S})^\top - \text{vec}(\mathbf{S}_p)\text{vec}(\mathbf{S}_p)^\top] \text{diag}(\mathbf{v}) \mathbf{W}_2^\top \mathbf{W}_2 \text{diag}(\mathbf{v})) \\
&= \text{tr}\left( (\text{vec}(\mathbf{S}_p)\text{vec}(\mathbf{S}_p)^\top + \text{diag}((\mathbb{E}[(1 - p_i + (2p_i - 1)d_i)^2 - (1 - p + (2p - 1)d_i)^2])_{i=1}^n)) \text{diag}(\mathbf{v}) \mathbf{W}_2^\top \mathbf{W}_2 \text{diag}(\mathbf{v}) \right) \\
&= \text{tr}\left( (\text{vec}(\mathbf{S}_p)\text{vec}(\mathbf{S}_p)^\top + \text{diag}((p(1-p)(1-2d_i)^2)_{i=1}^n)) \text{diag}(\mathbf{v}) \mathbf{W}_2^\top \mathbf{W}_2 \text{diag}(\mathbf{v}) \right) \\
&= \text{tr}\left( (\text{vec}(\mathbf{S}_p)\text{vec}(\mathbf{S}_p)^\top + p(1-p)(1-2\mathbf{D})^2) \text{diag}(\mathbf{v}) \mathbf{W}_2^\top \mathbf{W}_2 \text{diag}(\mathbf{v}) \right).
\end{aligned}
$$

Here, $p_i$ and $d_i$ are $(i, i)$ element of $\mathbf{P}$ and $\mathbf{D}$, respectively. Similarly, we expand the objective function when the activation function is modified leaky ReLU, $r_p(\cdot)$:

$$
\begin{aligned}
\| \mathbf{W}_2 r_p(\mathbf{W}_1 \mathbf{x}) - \mathbf{y} \|_2^2 &= \| \mathbf{W}_2 \left( (1-p)\mathbf{I} + (2p-1)\mathbf{D} \right)(\mathbf{W}_1 \mathbf{x}) - \mathbf{y} \|_2^2 \\
&= \| \mathbf{W}_2 \mathbf{S}_p \mathbf{v} - \mathbf{y} \|_2^2 \\
&= \text{tr}(\mathbf{W}_2 \mathbf{S}_p \mathbf{v} \mathbf{v}^\top \mathbf{S}_p \mathbf{W}_2^\top) - 2\text{tr}(\mathbf{W}_2 \mathbf{S}_p \mathbf{v} \mathbf{y}^\top) + \text{tr}(\mathbf{y} \mathbf{y}^\top)
\end{aligned}
\tag{5}
$$

$$\text{where, } \text{tr}(\mathbf{W}_2\mathbf{S}_p\mathbf{v}\mathbf{v}^\top\mathbf{S}_p\mathbf{W}_2^\top) = \text{tr}(\mathbf{S}_p\mathbf{v}\mathbf{v}^\top\mathbf{S}_p\mathbf{W}_2^\top\mathbf{W}_2)$$
$$= \text{tr}(\text{diag}(\mathbf{v})\text{vec}(\mathbf{S}_p)\text{vec}(\mathbf{S}_p)^\top\text{diag}(\mathbf{v})\mathbf{W}_2^\top\mathbf{W}_2)$$
$$= \text{tr}(\text{vec}(\mathbf{S}_p)\text{vec}(\mathbf{S}_p)^\top\text{diag}(\mathbf{v})\mathbf{W}_2^\top\mathbf{W}_2\text{diag}(\mathbf{v})).$$

To determine the relationship between the two objective functions (4) and (5), we subtract one equation from the other:

$$\mathbb{E}\big[\|\mathbf{W}_2\big(\mathbf{I} - \mathbf{P} + (2\mathbf{P} - \mathbf{I})r\big)(\mathbf{W}_1\mathbf{x}) - \mathbf{y}\|_2^2\big] - \|\mathbf{W}_2\big((1-p)\mathbf{I} + (2p-1)\mathbf{D}\big)(\mathbf{W}_1\mathbf{x}) - \mathbf{y}\|_2^2$$
$$= \mathbb{E}\big[\text{tr}(\mathbf{W}_2\mathbf{S}\mathbf{v}\mathbf{v}^\top\mathbf{S}\mathbf{W}_2^\top)\big] - \text{tr}(\mathbf{W}_2\mathbf{S}_p\mathbf{v}\mathbf{v}^\top\mathbf{S}_p\mathbf{W}_2^\top)$$
$$= \text{tr}\big(p(1-p)(1-2\mathbf{D})^2\text{diag}(\mathbf{v})\mathbf{W}_2^\top\mathbf{W}_2\text{diag}(\mathbf{v})\big)$$
$$= p(1-p)\text{tr}\big(\mathbf{W}_2\text{diag}(\mathbf{v})(\mathbf{I} - 2\mathbf{D})(\mathbf{I} - 2\mathbf{D})^\top\text{diag}(\mathbf{v})^\top\mathbf{W}_2^\top\big)$$
$$= p(1-p)\|\mathbf{W}_2(\mathbf{I} - 2\mathbf{D})\text{diag}(\mathbf{W}_1\mathbf{x})\|_2^2$$
$$\geq \frac{p(1-p)}{n}\|\mathbf{W}_2(\mathbf{I} - 2\mathbf{D})\mathbf{W}_1\mathbf{x}\|_2^2 \tag{6}$$
$$= \frac{p(1-p)}{n(2p-1)^2}\|\mathbf{W}_2\mathbf{W}_1\mathbf{x} - 2\mathbf{W}_2r_p(\mathbf{W}_1\mathbf{x})\|_2^2 \tag{7}$$
$$= \frac{4p(1-p)}{n(2p-1)^2}\left\|\mathbf{W}_2\left(\frac{1}{2}\mathbf{W}_1\mathbf{x}\right) - \mathbf{W}_2r_p(\mathbf{W}_1\mathbf{x})\right\|_2^2$$

To derive Equation (7), we utilize the definition of $\mathbf{S}_p$ introduced at the beginning of the proof:

$$\mathbf{S}_p = (1-p)\mathbf{I} + (2p-1)\mathbf{D}$$
$$\implies \mathbf{D} = \frac{1}{2p-1}\big(\mathbf{S}_p - (1-p)\mathbf{I}\big)$$
$$\implies \mathbf{I} - 2\mathbf{D} = \mathbf{I} - \frac{2}{2p-1}\big(\mathbf{S}_p - (1-p)\mathbf{I}\big)$$
$$= \frac{(2p-1)\mathbf{I} - 2\mathbf{S}_p + 2(1-p)\mathbf{I}}{2p-1}$$
$$= \frac{\mathbf{I} - 2\mathbf{S}_p}{2p-1}$$

Since $r_p(\cdot) = \mathbf{S}_p(\cdot)$, we get Equation (7). Finally, we get the relationship between these two objective functions:

$$\mathbb{E}\big[\|\mathbf{W}_2 AID_p(\mathbf{W}_1\mathbf{x}) - \mathbf{y}\|_2^2\big] \geq \|\mathbf{W}_2 r_p(\mathbf{W}_1\mathbf{x}) - \mathbf{y}\|_2^2$$
$$+ \frac{4p(1-p)}{n(2p-1)^2}\left\|\mathbf{W}_2\left(\frac{1}{2}\mathbf{W}_1\mathbf{x}\right) - \mathbf{W}_2 r_p(\mathbf{W}_1\mathbf{x})\right\|_2^2.$$

This means that objective function of AID gives the upper bound of the optimization problem when we use modified leaky ReLU and penalty term. The penalty term means that weight matrices $\mathbf{W}_1, \mathbf{W}_2$ are trained to closely resemble the behavior of the linear network. As $p$ approaches 0.5, the regularization effect becomes stronger, which intuitively aligns with the fact that AID is equivalent to a linear network when $p = 0.5$. Since $\big(\mathbb{E}[\|\mathbf{W}_2 AID_p(\mathbf{W}_1\mathbf{x}) - \mathbf{y}\|_2^2] \to 0\big) \implies \mathbf{y} = \mathbf{W}_2 r_p(\mathbf{W}_1\mathbf{x}) = \mathbf{W}_2\big(\frac{1}{2}\mathbf{W}_1\mathbf{x}\big)$, we confirm that $AID$ has a regularizing effect that constrains the network to behave like a linear network[2].

---

[2]We identified an error made by Liang et al. (2021) while deriving Equation (6), and by correcting it with an inequality, the claim becomes weaker compared to the original proof. However, we still assert that the regularization effect remains valid.

□

## C. Additional Studies

### C.1. Exploration of AID in Relation to ReLU, Dropout and DropReLU

We explore that when the number of the interval is two around zero (i.e., $k = 2$ and $u_1 = l_2 = 0$), AID encompasses several well-known concepts, including ReLU activation, Dropout, and DropReLU. This relationship is formalized in the following property:

*Property* 3. Let $AID_{p,q}$ be the function that dropout rate $p$ for positive and $q$ for negative values. Then, in training process, ReLU, Dropout, and DropReLU can be expressed by AID formulation.

For simplicity, without derivation, we describe the relationships between the equations below. We assume that Bernoulli sampling generates an identical mask within the same layer. Let $r(\cdot)$ be the ReLU function, $Dropout_p(\cdot)$ be the Dropout layer with probability $p$, and $DropReLU_p(\cdot)$ be an activation function that applies ReLU with probability $p$ and identity with probabiltiy $1 - p$. Then, during training process, for input vector $\mathbf{x}$, those functions can be represented by AID as below:

- $r(\mathbf{x}) = AID_{0,1}(\mathbf{x})$

- $Dropout_p(\mathbf{x}) = AID_{p,p}(\mathbf{x})/(1 - p)$

- $Dropout_p(r(\mathbf{x})) = AID_{p,1}(\mathbf{x})/(1 - p)$

- $DropReLU_p(\mathbf{x}) = AID_{0,p}(\mathbf{x})$

Property 3 indicates that using AID guarantees performance at least for the maximum performance of models using ReLU activation alone, in combination with dropout, or with DropReLU. Analysis and investigation of the AID function under varying hyperparameters such as intervals and probabilities are left as future work.

### C.2. Effect of Theorem 4.1 for ReLU and Standard Dropout

We analyze the changes in the formulation when applying a dropout layer to the ReLU function instead of AID. Our findings show that this approach does not achieve the same regularization effect toward a linear network as AID. We discuss this in Corollary C.1 below.

**Corollary C.1.** *Similar setting with Theorem 4.1, combination of ReLU activation and Dropout layer does not have effect that regularize to linear network.*

*Proof.* We use same notation with Theorem 4.1. We can write the prediction of the model that use Dropout layer with probability $p$ after ReLU activation, $\hat{\mathbf{y}}_i$ as below:

$$\hat{\mathbf{y}}_i = \mathbf{W}_2 \left( \frac{1}{1 - p}(\mathbf{I} - \mathbf{P})r(\mathbf{W}_1\mathbf{x}) \right)$$

where $r$ is ReLU function. On test phase, $\hat{\mathbf{y}}_i = \mathbf{W}_2 r(\mathbf{W}_1\mathbf{x})$. Then, to simplify the equations, let us denote $\mathbf{S}$, $\mathbf{S}_p$ and $\mathbf{v}$ as follows:

$$\mathbf{S} \doteq \frac{1}{1 - p}(\mathbf{I} - \mathbf{P})\mathbf{D}$$
$$\mathbf{S}_p \doteq \mathbf{D}$$
$$\mathbf{v} \doteq \mathbf{W}_1\mathbf{x}.$$

Then, following the same process as in Theorem 4.1, we derive the expansion for the relationship between $\mathbb{E}\left[ \|\mathbf{W}_2 \frac{1}{1-p}(\mathbf{I} - \mathbf{P})r(\mathbf{W}_1\mathbf{x}) - \mathbf{y}\|_2^2 \right]$ and $\|\mathbf{W}_2 r(\mathbf{W}_1\mathbf{x}) - \mathbf{y}\|_2^2$. We note that the detailed derivation is omitted as it largely overlaps with the previously established proof.

$$\mathbb{E}\left[\|\mathbf{W}_2\frac{1}{1-p}(\mathbf{I}-\mathbf{P})r(\mathbf{W}_1\mathbf{x})-\mathbf{y}\|_2^2\right]-\|\mathbf{W}_2r(\mathbf{W}_1\mathbf{x})-\mathbf{y}\|_2^2 = \mathrm{tr}\left(\frac{p}{1-p}\mathbf{D}^2\mathrm{diag}(\mathbf{v})\mathbf{W}_2^\top\mathbf{W}_2\mathrm{diag}(\mathbf{v})\right)$$

$$= \frac{p}{1-p}\mathrm{tr}(\mathbf{W}_2\mathrm{diag}(\mathbf{v})\mathbf{D}\mathbf{D}^\top\mathrm{diag}(\mathbf{v})^\top\mathbf{W}_2^\top)$$

$$= \frac{p}{1-p}\|\mathbf{W}_2\mathbf{D}\mathrm{diag}(\mathbf{v})\|_2^2$$

$$\geq \frac{p}{n(1-p)}\|\mathbf{W}_2r(\mathbf{W}_1\mathbf{x})\|_2^2$$

$$\implies \mathbb{E}\left[\|\mathbf{W}_2\frac{1}{1-p}(\mathbf{I}-\mathbf{P})r(\mathbf{W}_1\mathbf{x})-\mathbf{y}\|_2^2\right] \geq \|\mathbf{W}_2r(\mathbf{W}_1\mathbf{x})-\mathbf{y}\|_2^2 + \frac{p}{n(1-p)}\|\mathbf{W}_2r(\mathbf{W}_1\mathbf{x})\|_2^2.$$

$\square$

Corollary C.1 demonstrate that the simultaneous use of Dropout and ReLU does not achieve the same regularization effect on the linear network as AID. This suggests that, rather than completely discarding all negative values as ReLU does, allowing a portion of negative values to pass through with dropout plays a crucial role in effect that regularizes to linear network.

# D. Details of Property 2

At the forward pass case, the property of the ReLU function used in the derivation of equations in He et al. (2015)'s work as follows:

$$\mathbb{E}[(r(y))^2] = \frac{1}{2}\mathbb{E}[y^2]$$
$$= \frac{1}{2}(\mathbb{E}[y^2] - \mathbb{E}[y]^2)$$
$$= \frac{1}{2}Var(y)$$

where $r(\cdot)$ is ReLU activation, $y$ represents preactivation value which is a random variable from a zero-centered symmetric distribution. We show that substituting ReLU with AID in this equation yields the same result:

$$\mathbb{E}[(AID_p(y))^2] = p\mathbb{E}[(r(y))^2] + (1-p)\mathbb{E}[(\bar{r}(y))^2]$$
$$= p \times \frac{1}{2}Var(y) + (1-p) \times \frac{1}{2}Var(y)$$
$$= \frac{1}{2}Var(y).$$

Additionally, during the backward pass , He et al. (2015) utilize that derivative of ReLU function, $r'(y)$, is zero or one with the same probabilities. We have same condition that $y$ is a random variable from a zero-centered symmetric distribution. Applying this to AID, we can confirm that it yields the same result:

$$P(AID'_p(y) = 1) = P(AID'_p(y) = 1|y \geq 0)P(y \geq 0) + P(AID'_p(y) = 1|y < 0)P(y < 0)$$
$$= p \times \frac{1}{2} + (1-p) \times \frac{1}{2}$$
$$= \frac{1}{2}$$
$$P(AID'_p(y) = 0) = P(AID'_p(y) = 0|y \geq 0)P(y \geq 0) + P(AID'_p(y) = 0|y < 0)P(y < 0)$$
$$= (1-p) \times \frac{1}{2} + p \times \frac{1}{2}$$
$$= \frac{1}{2}.$$

Therefore, under the given conditions, replacing the ReLU function with AID during the derivation process yields identical results. Consequently, it can be concluded that Kaiming He initialization is also well-suited for AID.

# E. Baselines

**Dropout (Srivastava et al., 2014).** Dropout randomly deactivates a fraction of neurons during training, effectively simulating an ensemble of sub-networks by preventing specific pathways from dominating the learning process. This approach improves generalization by reducing overfitting and encouraging the model to learn more robust and diverse representations. We applied Dropout layer next to the activation functions.

**L2 Regularization (Krogh & Hertz, 1991).** Also known as weight decay, L2 regularization (L2) adds a penalty proportional to the squared magnitude of the model weights to the loss function. This encourages smaller weights, which can lead to better generalization.

**L2 Init Regularization (Kumar et al., 2023).** L2 Init introduces a regularization term that penalizes deviations of parameters from their initial values, aiming to preserve plasticity in continual learning. Unlike standard L2 regularization, which penalizes large weight magnitudes, it specifically focuses on maintaining the parameters near their initialization.

**Streaming Elastic Weight Consolidation (Elsayed & Mahmood, 2024).** Streaming Elastic Weight Consolidation (S-EWC) is a technique used in continual learning to prevent catastrophic forgetting. It identifies critical weights for previously learned tasks and penalizes changes to those weights during subsequent task learning.

**Shrink & Perturb (Ash & Adams, 2020).** Shrink & Perturb (S&P) is a method that combines weight shrinking, which reduces the magnitude of the weights to regularize the model, with perturbations that add noise to the weights. This approach is particularly effective in warm-start scenarios. As done in prior study (Lee et al., 2024b), we use a single hyperparameter to control both the noise intensity and the shrinkage strength. Specifically, given $\theta$ as the learnable parameter, $\theta_0$ as the initial learnable parameter, and $\lambda$ as the coefficient of S&P, then the applying S&P is defined as: $\theta \leftarrow (1 - \lambda)\theta + \lambda\theta_0$.

**DASH (Shin et al., 2024).** Direction-Aware SHrinking (DASH) selectively shrinks weights based on their cosine similarity with the loss gradient, effectively retaining meaningful features while reducing noise memorization. This approach improves training efficiency and maintains plasticity, ensuring better generalization under stationary data distributions. We applied DASH between the stages of training for generalizability experiments.

**ReDo (Sokar et al., 2023).** Recycling Dormant Neurons (ReDo) is a technique designed to address the issue of dormant neurons in neural networks by periodically reinitializing inactive neurons. This method helps maintain network expressivity and enhances performance by reactivating unused neurons during training. We applied ReDo at the end of each task.

**Continual Backprop (Dohare et al., 2024).** Continual Backpropagation (CBP) is a method that enhances network plasticity in continual learning by periodically reinitializing a fraction of low-utilization neurons. This targeted reinitialization preserves adaptability to new tasks, effectively mitigating the loss of plasticity while preserving previously learned knowledge.

**Concatenated ReLU (Abbas et al., 2023).** Concatenated ReLU (CReLU) is a variant of the ReLU activation function that combines the outputs of both the positive and negative regions of the input. This technique enhances plasticity on Streaming ALE environment.

**Randomized ReLU (Xu, 2015).** Randomized ReLU (RReLU) behaves like a standard ReLU for positive inputs, while allowing negative inputs to pass through with a randomly determined scaling factor. This stochasticity can help improve model generalization by introducing diverse non-linearities.

**DropReLU (Liang et al., 2021).** DropReLU operates by probabilistically switching the ReLU activation function to an linear function during training. This method effectively achieves generalization while remaining compatible with most architectures.

**Deep Fourier Feature (Lewandowski et al., 2024).** Deep Fourier Features (Fourier) effectively address plasticity loss by leveraging the concatenation of sine and cosine functions, which approximate the embedding of deep linear network properties within a model.

**Full Reset.** To compare performance against models trained from scratch, we included full reset as a baseline. At each stage, we reset the model's parameters to their initial values before continuing training.

For baselines like CReLU and Fourier, which inherently double the output dimensionality, we ensured fair comparisons during trainability experiments by matching their number of learnable parameters to that of the vanilla model. In all other cases, we halved the output dimensionality to approximately halve the total number of parameters, thereby eliminating any advantage arising solely from increased parameter counts.

# F. Experimental Details

## F.1. Loss of plasticity with Dropout

### F.1.1. TRAINABILITY FOR DROPOUT

We employed an 8-layer MLP featuring 512 hidden units and trained it on 1400 samples from the MNIST dataset. The model is trained for 50 different tasks, with each task running for 100 epochs. To evaluate subnetwork plasticity, we extracted 10 subnetworks at the conclusion of each task, training these on new tasks for an additional 100 epochs and then calculated the mean final accuracy. Adam optimizer was utilized for model optimization.

### F.1.2. WARM-START

We used the ResNet-18 architecture described in Appendix F.3. In the warm-start scenario, the model was pre-trained on 10% of the CIFAR100 dataset for 1,000 epochs and continued training on the full dataset for 100 epochs, with the optimizer reset at the start of full dataset training. In the cold-start scenario, the model was trained on the full dataset for 100 epochs. Adam optimizer with learning rate 0.001 was utilized.

## F.2. Trainability

**Permuted MNIST.** We followed the setup of (Dohare et al., 2024). It consists of a total of 800 tasks that 60,000 images are fed into models only once with 512 batch size. In the beginning of each task, the pixel of images are permuted arbitrarily. We trained fully connected neural networks with three hidden layers. Each hidden layer has 2,000 units and followed by ReLU activation function.

**Random Label MNIST.** We conducted a variant of (Kumar et al., 2023). It consists of a total of 200 tasks that 1,600 images are fed into models with 64 batch size. In the beginning of each task, the label class of images are changed to other class arbitrarily. We trained fully connected neural networks with three hidden layers 100 epochs per each task. Each hidden layer has 2,000 units and followed by ReLU activation function.

## F.3. Generalizability

We conducted experiments on CIFAR-10, CIFAR-100 (Krizhevsky et al., 2009), and TinyImageNet (Le & Yang, 2015) datasets using a 4-layer CNN, ResNet-18, and VGG-16, respectively, to evaluate the effectiveness of AID across different model architectures and datasets. We provide detailed model architecture below:

- **CNN**: We employed a convolutional neural network (CNN), which is used in relatively small image classification. The model includes two convolutional layers with a $5 \times 5$ kernel and 16 channels and max-pooling layer is followed after activation function. The two fully connected layers follow with 100 hidden units.

- **Resnet-18** (He et al., 2016): We utilized Resnet-18 to examine how well AID integrates with modern deep architectures featuring residual connections. Following (Lee et al., 2024b), the stem layers were removed to accommodate the smaller image size of the dataset. Additionally, a gradient clipping threshold of 0.5 was applied to ensure stable training.

- **VGG-16** (Simonyan & Zisserman, 2014): We adopted VGG-16 with batch normalization to investigate whether AID adapts properly in large-size models. The number of hidden units of the classifiers was set to 4096 without dropout.

In addition, we replaced the maxpooling layer with an average pooling layer for methods such as Fourier activation, DropReLU, and AID, where large values may not necessarily represent important features. Next, we provide the detailed experimental settings below.

**Continual Full.** Similar to the setup provided by (Shen et al., 2024), the entire data is randomly split into 10 chunks. The training process consists of 10 stages and the model gains access up to the $k$-th chunk in each stage $k$. In each stage, the dataset is fed into models with a batch size of 256. We trained the model for 100 epochs per each stage, and we reset the optimizer when each stage starts training.

**Continual Limited.** This setting follows the same configuration as continual full, with the key distinction that the model does not retain access to previously seen data chunks. At each stage, the model is trained only on the current chunk, without revisiting earlier data, simulating real-world constraints such as memory limitations and privacy concerns.

**Class-Incremental.** For CIFAR100 and TinyImageNet, the dataset was divided into 20 class-based chunks, with new classes introduced incrementally at each stage. Unless otherwise specified, test accuracy is evaluated based on the corresponding classes encountered up to each stage. The rest of the setup, including the batch size, and training epochs per stage, follows the Continual Full setting.

### F.4. Reinforcement Learning

To evaluate whether AID enhances sample efficiency on reinforcement learning, we conducted experiments on 17 Atari games from the Arcade Learning Environment (Bellemare et al., 2013), selected based on prior studies (Kumar et al., 2020; Sokar et al., 2023). We trained a DQN model following Mnih et al. (2015) using the CleanRL framework (Huang et al., 2022). The replay ratio was set to 1, as adopted in Sokar et al. (2023); Elsayed et al. (2024). We followed the hyperparameter settings for environment from Sokar et al. (2023), with details provided in Table 2.

*Table 2.* Hyperparameters used in reinforcement learning environment.

| Parameter | Value |
|---|---:|
| Optimizer | Adam (Kingma, 2014) |
| Optimizer: $\epsilon$ | $1.5\mathrm{e}-4$ |
| Optimizer: Learning rate | $6.25\mathrm{e}-4$ |
| Minimum $\epsilon$ for training | 0.01 |
| Evaluation $\epsilon$ | 0.001 |
| Discount factor $\gamma$ | 0.99 |
| Replay buffer size | $10^6$ |
| Minibatch size | 32 |
| Initial collect steps | 20000 |
| Training iterations | 10 |
| Training environment steps per iteration | $250K$ |
| Updates per environment step (Replay Ratio) | 1 |
| Target network update period | 2000 |
| Loss function | Huber Loss (Huber, 1992) |

### F.5. Standard Supervised Learning

For the same model architecture and dataset used in the generalizability experiments, we trained with Adam optimizer for 200 epochs, applying learning rate decay at the 100th and 150th epochs. The initial learning rate was set to 0.001 and was reduced by a factor of 10 at each decay step.

## F.6. Hyperparameter Search Space

We present the hyperparameter search space considered for each experiment in Table 3. Without mentioned, we performed a sweep over 5 different seeds for all experiments, except for VGG-16 model on the TinyImageNet dataset, where we used only 3 seeds due to computational cost. Tables 4 to 10 shows the best hyperparameter set that we found in various experiments.

*Table 3.* Hyperparameter search space for every experiment.

| Experiment | Method | Hyperparameters | Search Space |
|---|---|---|---|
| Warm-Start | Dropout | $p$ | $0.1, 0.3, 0.5$ |
| | AID | $p$ | $0.7, 0.8, 0.9$ |
| Trainability | Adam | learning rate | $1e-3, 1e-4$ |
| | SGD | learning rate | $3e-2, 3e-3$ |
| | L2 | $\lambda$ | $1e-2, 1e-3, 1e-4, 1e-5, 1e-6$ |
| | L2 Init | $\lambda$ | $1e-2, 1e-3, 1e-4, 1e-5, 1e-6$ |
| | Dropout | $p$ | $0.01, 0.05, 0.1, 0.15, 0.2, 0.25, 0.3$ |
| | S&P | $\lambda$ | $0.1, 0.2, 0.3, 0.4, 0.5, 0.6, 0.7, 0.8, 0.9$ |
| | ReDo | threshold ($\tau$) | $0.0, 1.0, 5.0, 10.0, 50.0$ |
| | CBP | replacement rate($\rho$) | $1e-1, 1e-2, 1e-3, 1e-4, 1e-5$ |
| | | maturity threshold | $100$ |
| | RReLU | lower | $0.0625, 0.125, 0.25$ |
| | | upper | $0.125, 0.25, 0.333, 0.5$ |
| | DropReLU | $p$ | $0.1, 0.2, 0.3, 0.4, 0.5, 0.6, 0.7, 0.8, 0.9, 0.99$ |
| | AID | $p$ | $0.1, 0.2, 0.3, 0.4, 0.5, 0.6, 0.7, 0.8, 0.9, 0.99$ |
| Generalizability | Adam | learning rate | $1e-3, 1e-4$ |
| | S&P | $\lambda$ | $0.2, 0.4, 0.6, 0.8$ |
| | CBP | replacement rate($\rho$) | $1e-4, 1e-5$ |
| | | maturity threshold | $100, 1000$ |
| | DropReLU | $p$ | $0.7, 0.8, 0.9$ |
| | L2 | $\lambda$ | $1e-2, 1e-3, 1e-4, 1e-5$ |
| | Dropout | $p$ | $0.1, 0.3, 0.5$ |
| | RReLU | lower | $0.125$ |
| | | upper | $0.333$ |
| | AID | $p$ | $0.7, 0.8, 0.9$ |
| Continual Full | DASH | $\alpha$ | $0.1, 0.3$ |
| | | $\lambda$ | $0.05, 0.1, 0.3$ |
| Class-Incremental | DASH | $\alpha$ | $0.1, 0.3$ |
| | | $\lambda$ | $0.05, 0.1, 0.3$ |
| Continual Limited | DASH | $\alpha$ | $0.1, 0.3, 0.7, 1.0$ |
| | | $\lambda$ | $0.3$ |
| | S-EWC | $\lambda$ | $100, 10, 1, 0.1, 0.01$ |
| Standard SL | L2 | $\lambda$ | $1e-2, 1e-3, 1e-4, 1e-5$ |
| | Dropout | $p$ | $0.1, 0.3, 0.5$ |
| | AID | $p$ | $0.8, 0.9, 0.95$ |
| Reinforcement Learning | Dropout | $p$ | $0.01, 0.001$ |
| | AID | $p$ | $0.99, 0.999$ |

*Table 4.* Hyperparameter set of each method on permuted MNIST.

| Method | Optimizer | Learning Rate | Optimal Hyperparameters |
|---|---|---|---|
| Baseline | Adam | 1e−3 | – |
| Dropout | Adam | 1e−3 | $p = 0.05$ |
| L2 | Adam | 1e−3 | $\lambda = 1\mathrm{e}{-5}$ |
| L2 Init | Adam | 1e−3 | $\lambda = 1\mathrm{e}{-4}$ |
| Shrink & Perturb | Adam | 1e−3 | $\lambda = 0.2$ |
| ReDo | Adam | 1e−3 | $\tau = 50$ |
| Continual Backprop | Adam | 1e−3 | $\rho = 1\mathrm{e}{-4}$ |
| CReLU | Adam | 1e−3 | – |
| RReLU | Adam | 1e−3 | $\mathrm{lower} = 0.0625, \mathrm{upper} = 0.333$ |
| DropReLU | Adam | 1e−3 | $p = 0.99$ |
| AID | Adam | 1e−3 | $p = 0.99$ |
| Baseline | Adam | 1e−4 | – |
| Dropout | Adam | 1e−4 | $p = 0.05$ |
| L2 | Adam | 1e−4 | $\lambda = 1\mathrm{e}{-5}$ |
| L2 Init | Adam | 1e−4 | $\lambda = 1\mathrm{e}{-4}$ |
| Shrink & Perturb | Adam | 1e−4 | $\lambda = 0.1$ |
| ReDo | Adam | 1e−4 | $\tau = 50$ |
| Continual Backprop | Adam | 1e−4 | $\rho = 1\mathrm{e}{-4}$ |
| CReLU | Adam | 1e−4 | – |
| RReLU | Adam | 1e−4 | $\mathrm{lower} = 0.0625, \mathrm{upper} = 0.333$ |
| DropReLU | Adam | 1e−4 | $p = 0.99$ |
| AID | Adam | 1e−4 | $p = 0.99$ |
| Baseline | SGD | 3e−2 | – |
| Dropout | SGD | 3e−2 | $p = 0.25$ |
| L2 | SGD | 3e−2 | $\lambda = 1\mathrm{e}{-5}$ |
| L2 Init | SGD | 3e−2 | $\lambda = 1\mathrm{e}{-3}$ |
| Shrink & Perturb | SGD | 3e−3 | $\lambda = 0.1$ |
| ReDo | SGD | 3e−2 | $\tau = 5$ |
| Continual Backprop | SGD | 3e−2 | $\rho = 1\mathrm{e}{-3}$ |
| CReLU | SGD | 3e−2 | – |
| RReLU | SGD | 3e−2 | $\mathrm{lower} = 0.0625, \mathrm{upper} = 0.333$ |
| DropReLU | SGD | 3e−2 | $p = 0.99$ |
| AID | SGD | 3e−2 | $p = 0.99$ |
| Baseline | SGD | 3e−3 | – |
| Dropout | SGD | 3e−3 | $p = 0.01$ |
| L2 | SGD | 3e−3 | $\lambda = 1\mathrm{e}{-5}$ |
| L2 Init | SGD | 3e−3 | $\lambda = 1\mathrm{e}{-3}$ |
| Shrink & Perturb | SGD | 3e−3 | $\lambda = 0.1$ |
| ReDo | SGD | 3e−3 | $\tau = 1$ |
| Continual Backprop | SGD | 3e−3 | $\rho = 1\mathrm{e}{-5}$ |
| CReLU | SGD | 3e−3 | – |
| RReLU | SGD | 3e−3 | $\mathrm{lower} = 0.0625, \mathrm{upper} = 0.333$ |
| DropReLU | SGD | 3e−3 | $p = 0.99$ |
| AID | SGD | 3e−3 | $p = 0.99$ |

*Table 5.* Hyperparameter set of each method on random label MNIST.

| Method | Optimizer | Learning Rate | Optimal Hyperparameters |
|---|---|---|---|
| Baseline | Adam | 1e−3 | – |
| Dropout | Adam | 1e−3 | $p = 0.15$ |
| L2 | Adam | 1e−3 | $\lambda = 1e{-}4$ |
| L2 Init | Adam | 1e−3 | $\lambda = 1e{-}3$ |
| Shrink & Perturb | Adam | 1e−3 | $\lambda = 0.8$ |
| ReDo | Adam | 1e−3 | $\tau = 0$ |
| Continual Backprop | Adam | 1e−3 | $\rho = 1e{-}3$ |
| CReLU | Adam | 1e−3 | – |
| RReLU | Adam | 1e−3 | $\text{lower} = 0.0625, \text{upper} = 0.125$ |
| DropReLU | Adam | 1e−3 | $p = 0.9$ |
| AID | Adam | 1e−3 | $p = 0.9$ |
| Baseline | Adam | 1e−4 | – |
| Dropout | Adam | 1e−4 | $p = 0.25$ |
| L2 | Adam | 1e−4 | $\lambda = 1e{-}3$ |
| L2 Init | Adam | 1e−4 | $\lambda = 1e{-}4$ |
| Shrink & Perturb | Adam | 1e−3 | $\lambda = 0.7$ |
| ReDo | Adam | 1e−4 | $\tau = 0$ |
| Continual Backprop | Adam | 1e−4 | $\rho = 1e{-}4$ |
| CReLU | Adam | 1e−4 | – |
| RReLU | Adam | 1e−4 | $\text{lower} = 0.25, \text{upper} = 0.333$ |
| DropReLU | Adam | 1e−4 | $p = 0.99$ |
| AID | Adam | 1e−4 | $p = 0.99$ |
| Baseline | SGD | 3e−2 | – |
| Dropout | SGD | 3e−2 | $p = 0.15$ |
| L2 | SGD | 3e−2 | $\lambda = 1e{-}3$ |
| L2 Init | SGD | 3e−2 | $\lambda = 1e{-}3$ |
| Shrink & Perturb | SGD | 3e−3 | $\lambda = 0.2$ |
| ReDo | SGD | 3e−2 | $\tau = 1$ |
| Continual Backprop | SGD | 3e−2 | $\rho = 1e{-}5$ |
| CReLU | SGD | 3e−2 | – |
| RReLU | SGD | 3e−2 | $\text{lower} = 0.0625, \text{upper} = 0.125$ |
| DropReLU | SGD | 3e−2 | $p = 0.99$ |
| AID | SGD | 3e−2 | $p = 0.99$ |
| Baseline | SGD | 3e−3 | – |
| Dropout | SGD | 3e−3 | $p = 0.01$ |
| L2 | SGD | 3e−3 | $\lambda = 1e{-}2$ |
| L2 Init | SGD | 3e−3 | $\lambda = 1e{-}3$ |
| Shrink & Perturb | SGD | 3e−3 | $\lambda = 0.1$ |
| ReDo | SGD | 3e−3 | $\tau = 1$ |
| Continual Backprop | SGD | 3e−3 | $\rho = 1e{-}3$ |
| CReLU | SGD | 3e−3 | – |
| RReLU | SGD | 3e−3 | $\text{lower} = 0.0625, \text{upper} = 0.25$ |
| DropReLU | SGD | 3e−3 | $p = 0.99$ |
| AID | SGD | 3e−3 | $p = 0.99$ |

*Table 6.* Hyperparameter set of each method on continual full setting.

| Dataset (Model) | Method | Optimal Learning Rate | Optimal Hyperparameters |
|---|---|---|---|
| CIFAR10 | Baseline | 1e−4 | – |
| (CNN) | Full Reset | 1e−4 | – |
| | L2 | 1e−3 | $\lambda = 1e{-}2$ |
| | Dropout | 1e−4 | $p = 0.3$ |
| | RReLU | 1e−4 | – |
| | CReLU | 1e−4 | – |
| | Fourier | 1e−4 | – |
| | S&P | 1e−4 | $\lambda = 0.8$ |
| | CBP | 1e−4 | $\rho = 1e{-}5$, maturity threshold $= 100$ |
| | DASH | 1e−4 | $\alpha = 0.1, \lambda = 0.3$ |
| | DropReLU | 1e−4 | $p = 0.9$ |
| | AID | 1e−3 | $p = 0.9$ |
| CIFAR100 | Baseline | 1e−3 | – |
| (Resnet-18) | Full Reset | 1e−3 | – |
| | L2 | 1e−3 | $\lambda = 1e{-}2$ |
| | Dropout | 1e−4 | $p = 0.3$ |
| | RReLU | 1e−3 | – |
| | CReLU | 1e−3 | – |
| | Fourier | 1e−3 | – |
| | S&P | 1e−3 | $\lambda = 0.8$ |
| | CBP | 1e−3 | $\rho = 1e{-}5$, maturity threshold $= 1000$ |
| | DASH | 1e−4 | $\alpha = 0.1, \lambda = 0.05$ |
| | DropReLU | 1e−3 | $p = 0.8$ |
| | AID | 1e−3 | $p = 0.7$ |
| TinyImageNet | Baseline | 1e−3 | – |
| (VGG-16) | Full Reset | 1e−3 | – |
| | L2 | 1e−3 | $\lambda = 1e{-}4$ |
| | Dropout | 1e−4 | $p = 0.1$ |
| | RReLU | 1e−4 | – |
| | CReLU | 1e−3 | – |
| | Fourier | 1e−4 | – |
| | S&P | 1e−3 | $\lambda = 0.8$ |
| | CBP | 1e−3 | $\rho = 1e{-}4$, maturity threshold $= 100$ |
| | DASH | 1e−4 | $\alpha = 0.3, \lambda = 0.1$ |
| | DropReLU | 1e−4 | $p = 0.7$ |
| | AID | 1e−4 | $p = 0.7$ |

*Table 7.* Hyperparameter set of each method on continual limited setting.

| Dataset (Model) | Method | Optimal Learning Rate | Optimal Hyperparameters |
|---|---|---|---|
| CIFAR10 | Baseline | 1e−4 | – |
| (CNN) | L2 | 1e−4 | $\lambda = 1e-5$ |
| | Dropout | 1e−3 | $p = 0.5$ |
| | S-EWC | 1e−4 | $\lambda = 0.01$ |
| | RReLU | 1e−4 | – |
| | CReLU | 1e−4 | – |
| | Fourier | 1e−4 | – |
| | S&P | 1e−4 | $\lambda = 0.2$ |
| | CBP | 1e−4 | $\rho = 1e-4$, maturity threshold $= 100$ |
| | DASH | 1e−3 | $\alpha = 0.1, \lambda = 0.3$ |
| | DropReLU | 1e−4 | $p = 0.9$ |
| | AID | 1e−3 | $p = 0.8$ |
| CIFAR100 | Baseline | 1e−3 | – |
| (Resnet-18) | L2 | 1e−3 | $\lambda = 1e-5$ |
| | Dropout | 1e−4 | $p = 0.1$ |
| | S-EWC | 1e−3 | $\lambda = 1$ |
| | RReLU | 1e−3 | – |
| | CReLU | 1e−3 | – |
| | Fourier | 1e−3 | – |
| | S&P | 1e−3 | $\lambda = 0.2$ |
| | CBP | 1e−3 | $\rho = 1e-5$, maturity threshold $= 1000$ |
| | DASH | 1e−4 | $\alpha = 0.1, \lambda = 0.3$ |
| | DropReLU | 1e−4 | $p = 0.7$ |
| | AID | 1e−3 | $p = 0.8$ |
| TinyImageNet | Baseline | 1e−4 | – |
| (VGG-16) | L2 | 1e−3 | $\lambda = 1e-4$ |
| | Dropout | 1e−4 | $p = 0.1$ |
| | S-EWC | 1e−3 | $\lambda = 100$ |
| | RReLU | 1e−4 | – |
| | CReLU | 1e−3 | – |
| | Fourier | 1e−4 | – |
| | S&P | 1e−3 | $\lambda = 0.4$ |
| | CBP | 1e−4 | $\rho = 1e-4$, maturity threshold $= 1000$ |
| | DASH | 1e−3 | $\alpha = 0.1, \lambda = 0.3$ |
| | DropReLU | 1e−4 | $p = 0.8$ |
| | AID | 1e−4 | $p = 0.8$ |

*Table 8.* Hyperparameter set of each method on class-incremental setting.

| Dataset (Model) | Method | Optimal Learning Rate | Optimal Hyperparameters |
|---|---|---|---|
| CIFAR100 | Baseline | 1e−3 | – |
| (Resnet-18) | Full Reset | 1e−3 | – |
| | L2 | 1e−3 | $\lambda = 1e{-}5$ |
| | Dropout | 1e−3 | $p = 0.3$ |
| | RReLU | 1e−3 | – |
| | CReLU | 1e−3 | – |
| | Fourier | 1e−3 | – |
| | S&P | 1e−3 | $\lambda = 0.4$ |
| | CBP | 1e−3 | $\rho = 1e{-}4$, maturity threshold $= 1000$ |
| | DASH | 1e−4 | $\alpha = 0.1, \lambda = 0.05$ |
| | DropReLU | 1e−4 | $p = 0.8$ |
| | AID | 1e−3 | $p = 0.7$ |
| TinyImageNet | Baseline | 1e−3 | – |
| (VGG-16) | Full Reset | 1e−3 | – |
| | L2 | 1e−3 | $\lambda = 1e{-}5$ |
| | Dropout | 1e−4 | $p = 0.1$ |
| | RReLU | 1e−4 | – |
| | CReLU | 1e−3 | – |
| | Fourier | 1e−4 | – |
| | S&P | 1e−3 | $\lambda = 0.4$ |
| | CBP | 1e−3 | $\rho = 1e{-}5$, maturity threshold $= 1000$ |
| | DASH | 1e−3 | $\alpha = 0.3, \lambda = 0.3$ |
| | DropReLU | 1e−4 | $p = 0.8$ |
| | AID | 1e−4 | $p = 0.7$ |

*Table 9.* Hyperparameter set of each method on reinforcement learning setting.

| Game | Dropout ($p$) | AID ($p$) |
|------|------|------|
| Seaquest | 0.01 | 0.999 |
| DemonAttack | 0.01 | 0.99 |
| SpaceInvaders | 0.01 | 0.99 |
| Qbert | 0.01 | 0.999 |
| DoubleDunk | 0.01 | 0.99 |
| MsPacman | 0.01 | 0.999 |
| Enduro | 0.001 | 0.99 |
| BeamRider | 0.01 | 0.99 |
| WizardOfWor | 0.01 | 0.999 |
| Jamesbond | 0.01 | 0.99 |
| RoadRunner | 0.01 | 0.999 |
| Asterix | 0.01 | 0.99 |
| Pong | 0.01 | 0.999 |
| Zaxxon | 0.01 | 0.999 |
| YarsRevenge | 0.01 | 0.99 |
| Breakout | 0.01 | 0.99 |
| IceHockey | 0.01 | 0.99 |

*Table 10.* Hyperparameter set of each method on standard supervised learning setting.

| Dataset(Model) | Method | Optimal Hyperparameters |
|------|------|------|
| CIFAR10 | L2 | $\lambda = 1e{-}2$ |
| (CNN) | Dropout | $p = 0.3$ |
|  | AID | $p = 0.95$ |
| CIFAR100 | L2 | $\lambda = 1e{-}5$ |
| (Resnet-18) | Dropout | $p = 0.1$ |
|  | AID | $p = 0.8$ |
| TinyImageNet | L2 | $\lambda = 1e{-}5$ |
| (VGG-16) | Dropout | $p = 0.1$ |
|  | AID | $p = 0.9$ |

# G. Omitted Results

## G.1. Additional Results for Trainability

### G.1.1. EXTENDED RESULTS FOR VARIOUS CONDITIONS

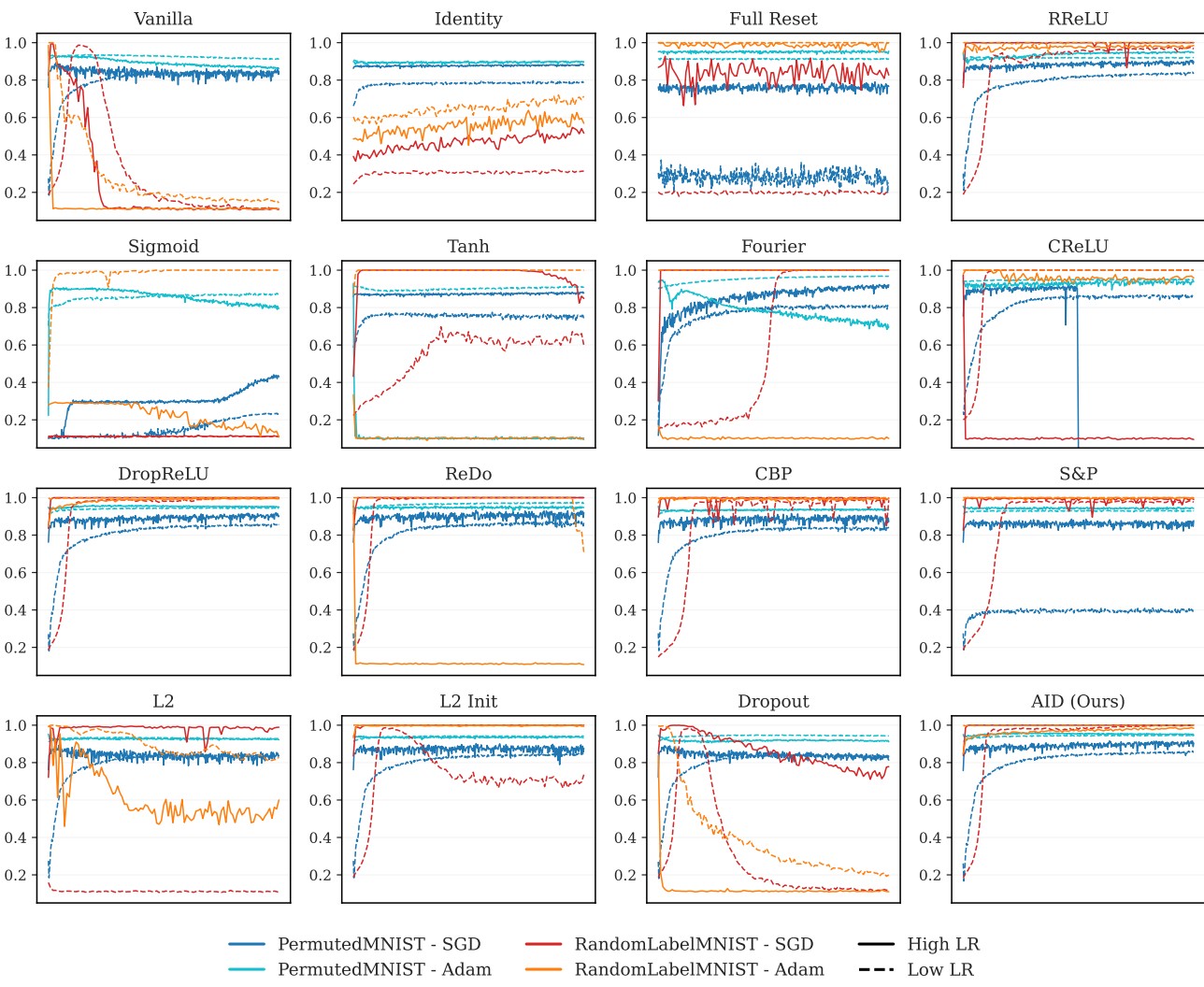

*Figure 7.* **Additional Results for Trainability on Various Conditions.** We plot the train accuracy comparisons across different optimizers and learning rates on Permuted MNIST and Random Label MNIST for each method. We train SGD with learning rate $3e-2, 3e-3$ and Adam with learning rate $1e-3, 1e-4$.

We conducted additional trainability experiments on various activation functions, including Identity, RReLU, Sigmoid, and Tanh, as presented in Figure 7. As discussed in Dohare et al. (2024); Lewandowski et al. (2024), linear networks do not suffer from plasticity loss. Interestingly, stochastic activation functions such as RReLU, DropReLU, and AID demonstrated strong trainability across different settings. This observation suggests that studying the properties of these activation functions may contribute to preserving plasticity, providing a promising direction for future research.

G.1.2. ANALYSIS OF METRICS CONTRIBUTING TO PLASTICITY LOSS

The precise cause of plasticity loss remains unclear, but several plausible indicators have been proposed to assess its impact. Among them, the dormant neuron ratio (Sokar et al., 2023) and effective rank (Kumar et al., 2020; Lyle et al., 2022) have been widely studied as key metrics. Recently, average sign entropy (Lewandowski et al., 2024) of preactivation values has been introduced as an additional measure. Each metric is computed as follows:

- **Dormant Neuron Ratio.** Sokar et al. (2023) introduced dormant neurons as a measure for the reduced expressivity of the neural network. A neuron is considered $\tau$-dormant, if its normalized activation score is lower than $\tau$. The normalized activation score is computed as follow:

$$s_i^l = \frac{\mathbb{E}_{\mathbf{x} \in D} \left[ |h_i^l(\mathbf{x})| \right]}{\frac{1}{H^l} \sum_{k \in [H^l]} \mathbb{E}_{\mathbf{x} \in D} \left[ |h_k^l(\mathbf{x})| \right]}$$

  where $s_i^l$ is normalized activation score for $i$-th neuron in $l$-th layer. $\mathbf{x} \in D$ is sample from input distribution, $H^l$ is the number of neurons in $l$-th layer, and $h_i^l(\cdot)$ is post-activation function. Note that $l$ does not include final layer of the network. The dormant neuron ratio is computed as the proportion of neurons with $\tau = 0$.

- **Average Unit Sign Entropy.** Lewandowski et al. (2024) proposed unit sign entropy as a generalized metric encompassing unit saturation (Abbas et al., 2023) and unit linearization (Lyle et al., 2024). It is defined as follows:

$$\text{Unit Sign Entropy}(x) = \mathbb{E}_{p(x)}[sgn(h(x))]$$

  where $p(x)$ is a distribution of inputs to the network, $h(\cdot)$ is preactivation values of according unit, and $sgn(\cdot)$ is sign function. To obtain the final metric, we compute the mean across all units, yielding the average unit sign entropy.

- **Effective Rank.** Previous studies (Kumar et al., 2020; Lyle et al., 2022) present the effective rank of the output matrix after the penultimate layer is closely related to plasticity loss. While several methods exist for computing effective rank, we follow srank (Kumar et al., 2020):

$$\text{srank}_\delta(\Phi) = \min_k \frac{\sum_{i=1}^k \sigma_i(\Phi)}{\sum_{j=1}^n \sigma_j(\Phi)} \geq 1 - \delta$$

  where $\Phi$ is feature matrix, and $\{\sigma_i(\Phi)\}_{i=1}^n$ is singular values of $\Phi$ sorted in descending order. We set $\delta = 0.01$ as default.

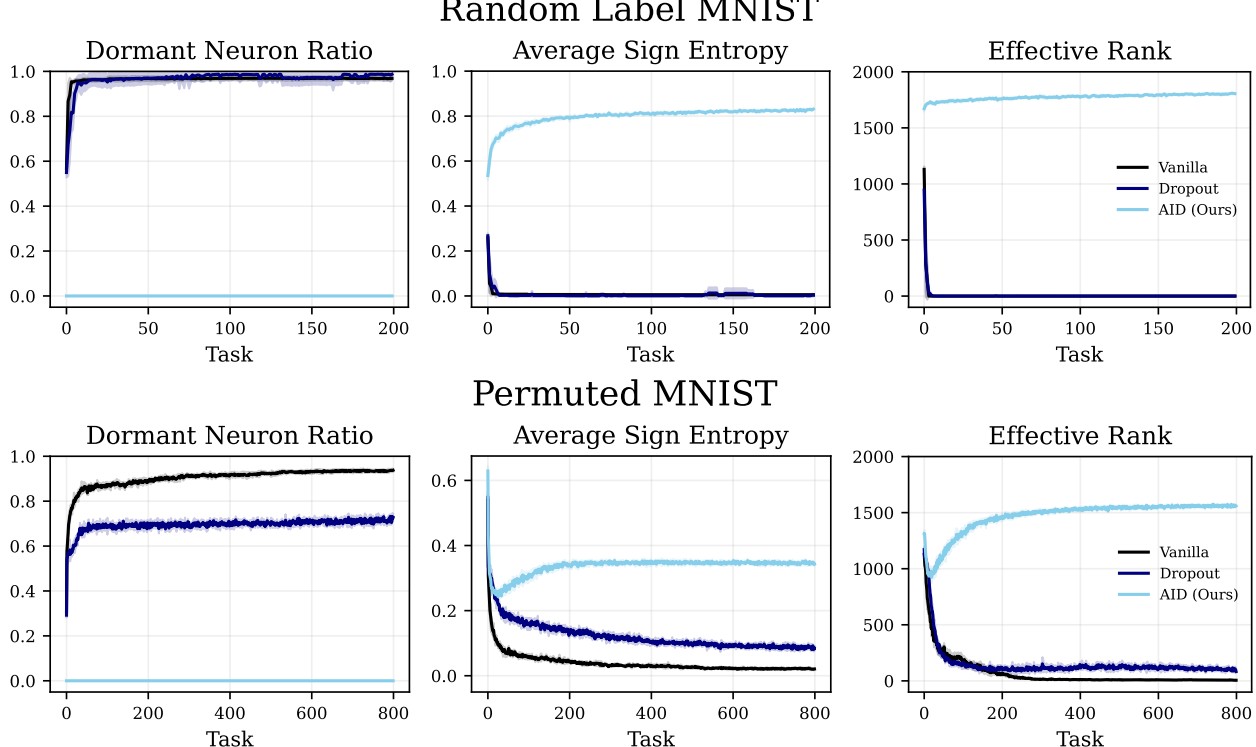

*Figure 8.* **Metrics for Measuring Plasticity Loss.** This figure presents the Dormant Neuron Ratio, Average Sign Entropy, and Effective Rank across tasks, comparing Vanilla, Dropout, and AID on random label MNIST (**Top**) and permuted MNIST (**Bottom**). Notably, AID maintains key metrics where Dropout fails, demonstrating its ability to mitigate plasticity loss and sustain network adaptability.

We evaluate these metrics for vanilla, Dropout, and AID under experimental settings characterized by severe plasticity loss. Specifically, we analyze permuted MNIST and random label MNIST with the Adam optimizer (lr = 0.001). The results are presented in Figure 8. Interestingly, Dropout had no effect on random label MNIST and provided only a slight improvement in permuted MNIST. While both Dropout and Vanilla models cause a significant portion of neurons to become dormant, AID effectively prevents neuron saturation. The Average Sign Entropy further supports this observation, as AID maintains consistently higher entropy, indicating a more diverse and active neuron distribution. Lastly, the Effective Rank plot demonstrates that AID preserves representational capacity throughout training, whereas both Dropout and Vanilla models experience a rapid decline. These findings highlight AID's ability to mitigate plasticity loss and sustain model adaptability over training.

### G.1.3. ANALYSIS ON PREACTIVATION DISTRIBUTION SHIFT OF DROPOUT AND AID

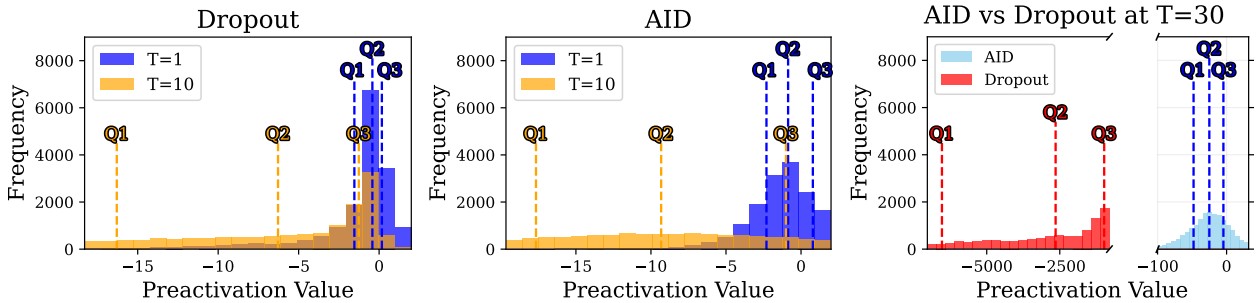

*Figure 9.* **Left & Middle.** Visualization of the preactivation distribution shift for Dropout and AID at the first and tenth tasks. Q1, Q2, and Q3 represent the first, second (median), and third quartiles, respectively. **Right.** Comparison of the preactivation distributions of AID and Dropout at the 30th task.

In this section, we present additional experimental results related to Section 3. Specifically, we analyze the preactivation distribution of Dropout and AID. The results presented in this section follow the Random label MNIST experimental setup as those in Section 3, with further details provided in Appendix F.1.1. We trained an 8-layer MLP on the random label MNIST and visualized the preactivation of the penultimate layer in Figure 9.

Figure 9 (left and middle) illustrate the preactivation distributions of Dropout and AID, for the first and tenth tasks, respectively. According to prior research (Lyle et al., 2024), preactivation distribution shift is one of the primary causes of plasticity loss in non-stationary settings. Dropout, which suffers from plasticity loss, exhibits a drastic preactivation shift, aligning well with previous findings. However, AID, despite maintaining plasticity effectively, undergoes a similarly significant preactivation shift as Dropout. This observation contradicts existing results, which indicate that the initial preactivation shift during training may not necessarily be the primary cause of plasticity loss.

Although both Dropout and AID experience preactivation distribution shifts, the extent of the shift in Dropout becomes more severe over the course of training compared to AID. Figure 9 (right) compares the preactivation distributions of Dropout and AID in the 30th task. As training progresses, Dropout's distribution shifts considerably, with the Q2 value approaching -2500, whereas AID maintains a distribution much closer to zero. These results indicate that while both Dropout and AID undergo distribution shifts, AID exhibits a bounded shift, preventing excessive plasticity loss. In contrast, Dropout experiences an unbounded distribution shift, leading to substantial plasticity degradation. This finding suggests that simply applying different Dropout probabilities across different stages, as in AID, can help mitigate preactivation distribution shifts.

### G.2. Additional Results for Generalizability

#### G.2.1. GENERALIZABILITY COMPARISON: DROPOUT VS. AID

To further validate the findings in Section 3.2, we conduct an extended comparison between Dropout and AID in terms of generalizability under the continual full setting. While Dropout is known to improve generalization performance, our results suggest that it does not enhance generalizability. To strengthen this conclusion, we perform a controlled experiment using the same model architecture, dataset, and learning rate search range.

For each method (Dropout and AID), we evaluate two variants: one where the model is reset after each data increment and one where it is not. For both settings, we identify the hyperparameter that yields the highest final accuracy at the last epoch. We then compute the generalizability gap by subtracting the accuracy of the non-resetting variant from that of the resetting one (i.e., (Cold-Start Acc.) - (Warm-Start Acc.)). A smaller value indicates better generalizability, as it implies that the method retains performance even without full retraining.

Figure 10 demonstrates that AID consistently maintains a lower generalizability gap across training epochs, indicating its superior generalizability. While Dropout exhibits a smaller gap compared to the vanilla model, it fails to prevent the gradual increase over time. Investigating the theoretical foundations behind AID's ability to preserve generalizability would be a valuable direction for future research in this area.

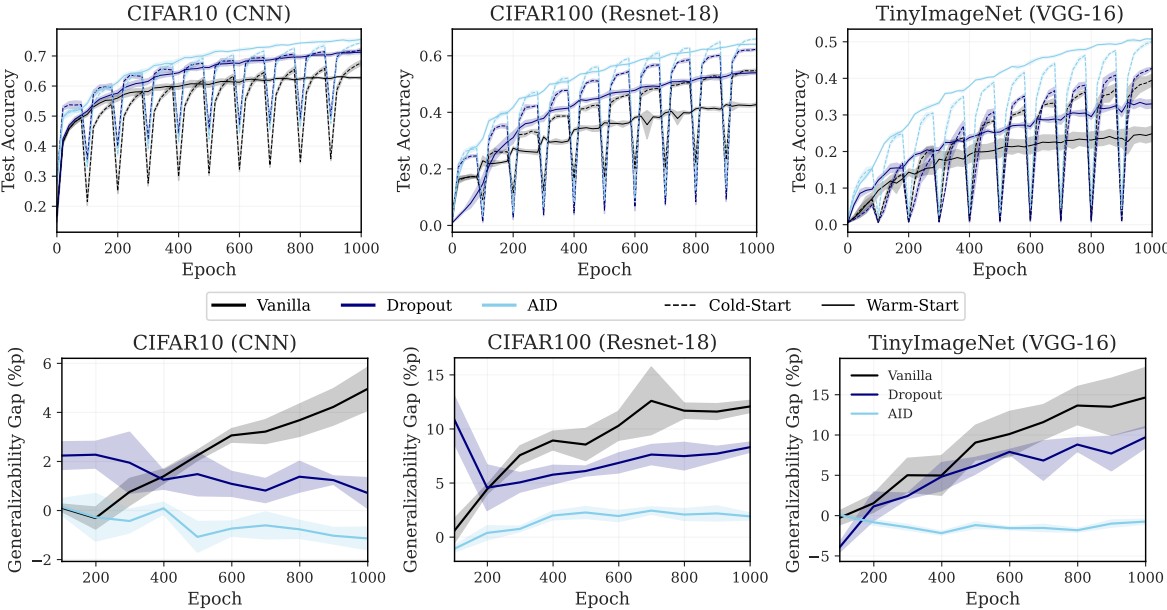

*Figure 10.* **Generalizability Comparison.** Test accuracies of cold-start and warm-start variants for vanilla, Dropout, and AID (**top row**). To facilitate direct comparison, we report the generalizability gap, computed as the accuracy difference between cold-start and warm-start (**bottom row**).

### G.2.2. HYPERPARAMETER SENSITIVITY

In this section, we present an additional study on the hyperparameter sensitivity of AID in the most representative setting, the continual full setup described in Section 5.1.2. We evaluate the effect of varying the hyperparameter $p$ in AID using the same models and datasets as in the main experiments. Specifically, we report the test accuracy at the final epoch for $p$ values in $[0.6, 0.7, 0.8, 0.9]$, and learning rates $0.001$ and $0.0001$.

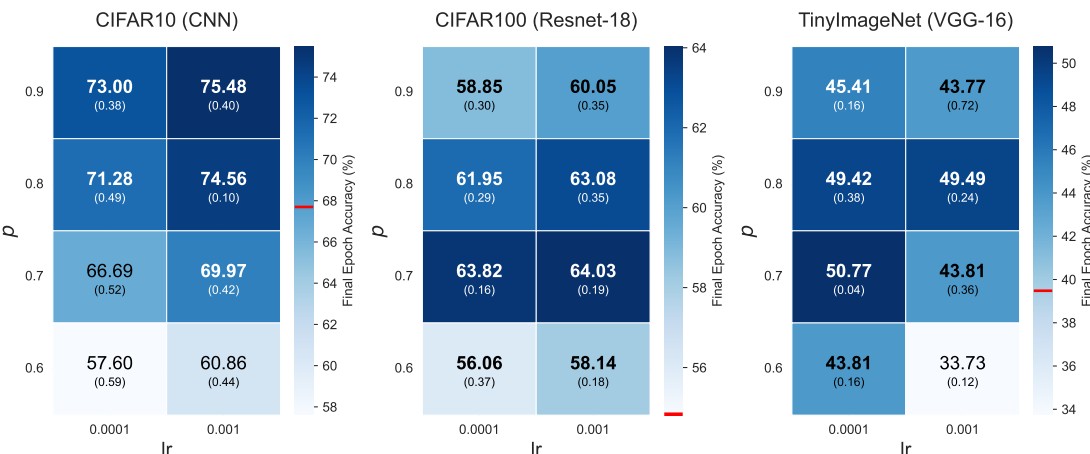

*Figure 11.* **Hyperparameter Sensitivity for AID.** Heatmaps show the final epoch test accuracy and standard deviation for different values of $p$ and learning rate on continual full setting. The red lines in the colorbars indicate the baseline accuracy of *full reset*. Values above this threshold are displayed in bold. Standard deviation is shown in smaller font below the mean accuracy.

The results are visualized as heatmaps in Figure 11, where each cell shows the mean accuracy (with standard deviation in smaller font). The red line on each colorbar indicates the baseline performance of the *full reset* model. Accuracy values above this baseline are highlighted in bold. Overall, AID achieves higher performance than the *full reset* baseline across most hyperparameter configurations, demonstrating the robustness of AID to the choice of $p$ and learning rate.

### G.3. Additional Results for Class-Incremental

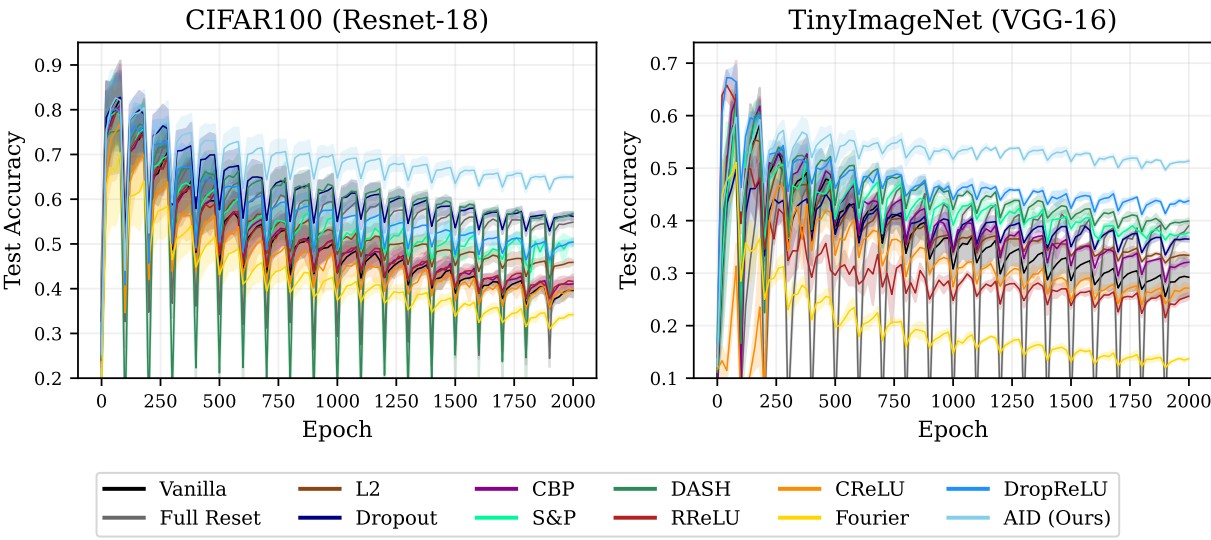

*Figure 12.* **Test Accuracy for Learned Classes in Class-Incremental Setting.** As new classes are introduced, the task complexity increases, leading to a gradual decline in accuracy.

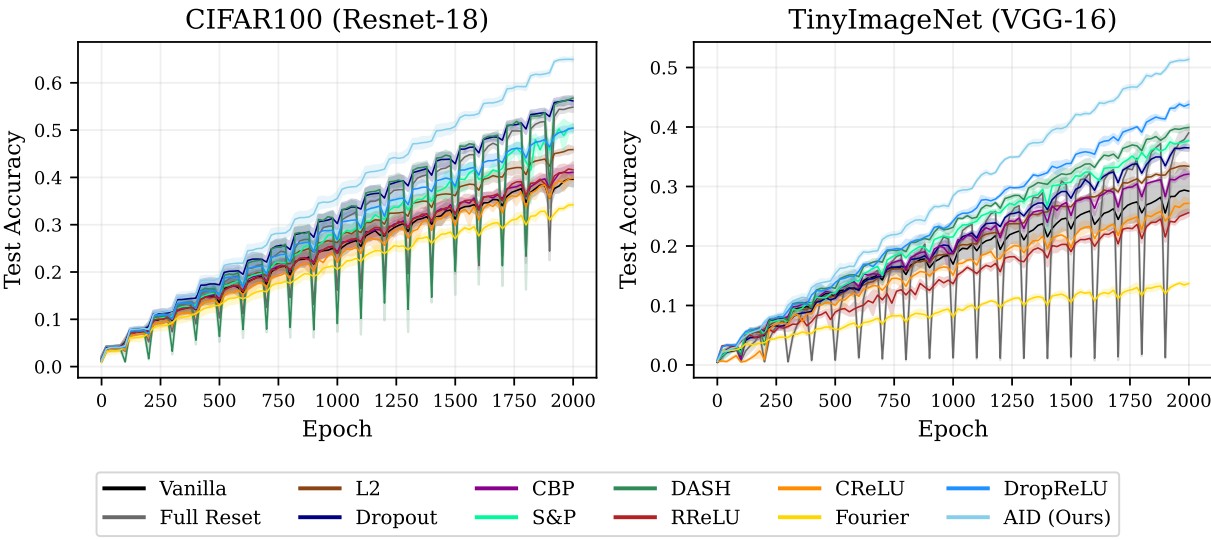

*Figure 13.* **Test Accuracy for Total Classes in Class-Incremental Setting.**

In the main section of the paper, we primarily presented results for the difference from full reset in class-incremental experiment. To provide a more detailed analysis, we include additional plots in Figures 12 and 13, illustrating both the accuracy for previously learned classes and the accuracy across all classes. These results further highlight that AID consistently outperforms the other proposed approaches, demonstrating its effectiveness in maintaining plasticity while mitigating performance degradation in class-incremental learning.

## G.4. Additional Results for Reinforcement Learning

### G.4.1. ATARI 2600 RAW SCORES BY GAMES

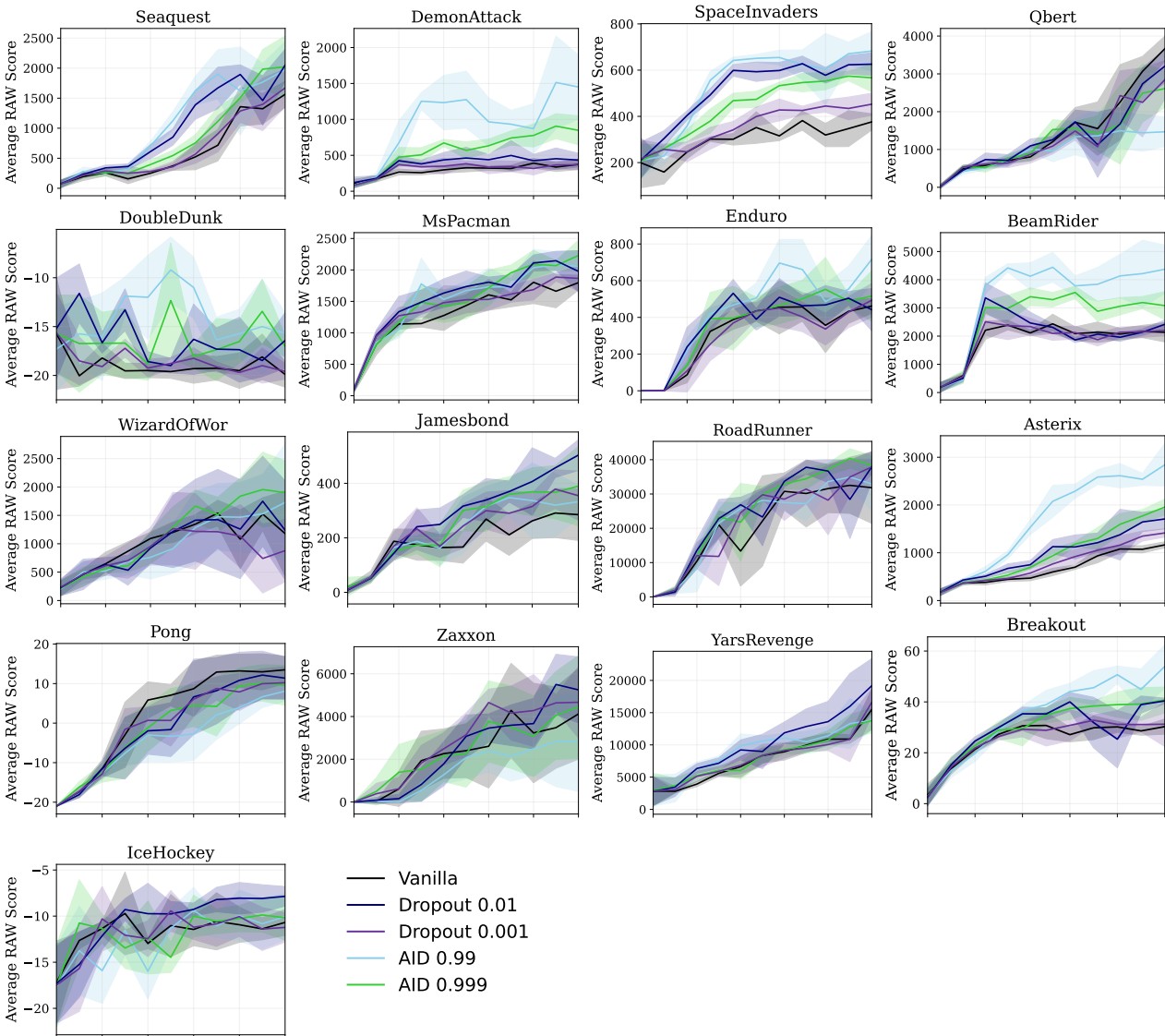

*Figure 14.* **Average RAW Scores across 17 Atari Games.** We train DQN model over 10 million frames with replay ratio 1. The comparison includes vanilla DQN, Dropout and AID. Shaded regions indicate the standard deviation across 5 runs.

In the reinforcement learning experiments, we sweep over relatively high coefficient for AID [0.99, 0.999], consistent with previous trainability experiments where a high coefficient was found to be optimal for continuously reducing training loss. For Dropout, we sweep over smaller coefficients [0.01, 0.001], since Dropout can cause large variance in reinforcement learning. Despite its lower trainability in some games, Dropout showed modest improvements in a few cases, possibly due to its model ensemble effect. However, in three games—Asterix, BeamRider and DemonAttack—which are known to suffer from plasticity loss (Sokar et al., 2023), Dropout failed to improve performance, supporting our argument that Dropout is ineffective against plasticity loss. While AID demonstrated advantages in most games, we observed a slight performance drop in certain cases, such as Pong and Qbert, where plasticity loss was not a significant issue even at high replay ratios. Further investigation is needed to understand this behavior, including extended training on these specific games. Additionally, our experiments were limited to high replay ratios; future research could explore alternative RL algorithms, different environments, or long-term learning settings to gain further insights.

G.4.2. IMPACT OF AID ON EFFECTIVE RANK

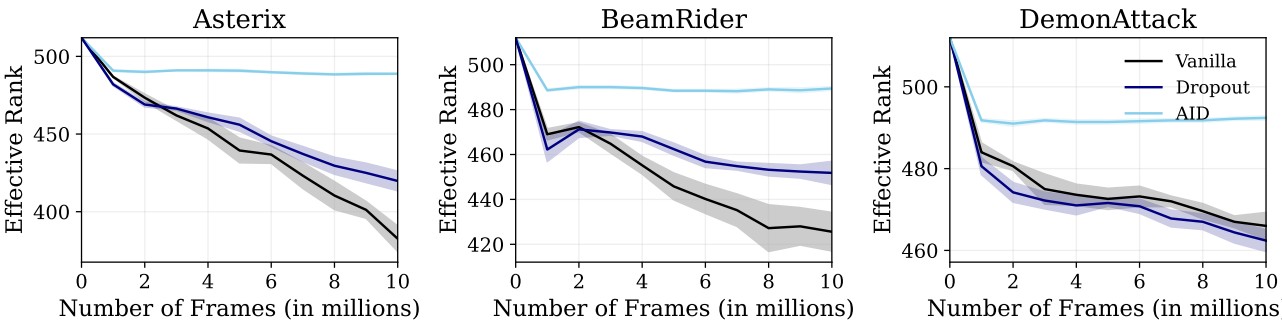

*Figure 15.* **Effective Rank of 3 Games.** The AID method maintains a higher effective rank compared to the vanilla and Dropout model throughout training.

Following the approach in Section G.1.2, we analyze the feature rank for three games—Asterix, BeamRider, and DemonAttack—where AID demonstrated significant performance improvements. As shown in Figure 15, the feature rank of the vanilla and Dropout model steadily declines throughout training, indicating a loss of representational diversity. In contrast, AID consistently maintains a higher feature rank, suggesting that it helps preserve network plasticity.

## G.5. Learning Curves for Standard Supervised Learning

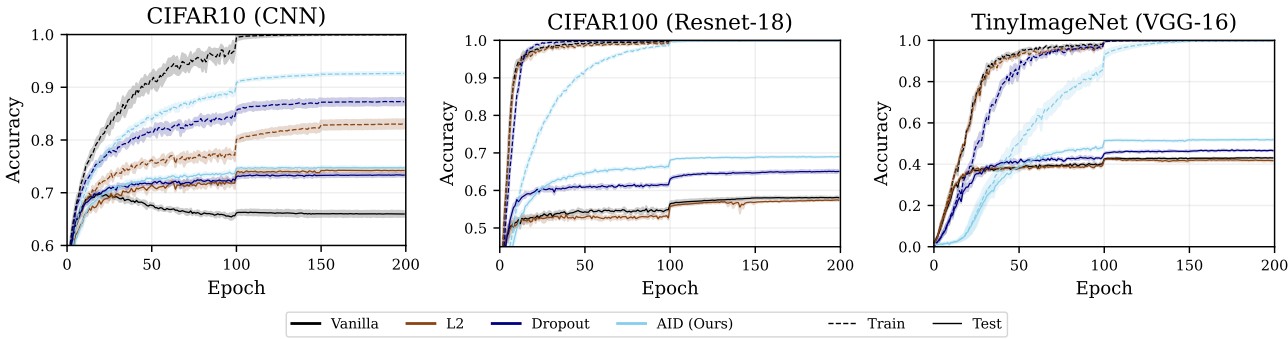

*Figure 16.* **Learning Curves on Standard Supervised Learning.** The figure presents training and test accuracy curves for CIFAR10 (CNN), CIFAR100 (Resnet-18), and TinyImageNet (VGG-16). We divide learning rate by 10 at 100th and 150th epochs. AID not only effectively mitigates plasticity loss but also reduces the generalization gap, demonstrating its robustness in both continual and standard supervised learning settings.

The Figure 16 present training and test loss of experiment in Section 5.3. We observe that AID more effectively closes the generalization gap compared to L2 regularization and Dropout, both of which are commonly used to address overfitting. This result suggests that AID is not merely a technique for mitigating plasticity loss but a more general-purpose methodology that enhances model robustness across various tasks.

# H. Implementation Details of AID

```python
import torch
import torch.nn as nn

class AID(nn.Module):
    def __init__(self, p=0.9):
        super(AID, self).__init__()
        self.p = float(p)

        if p < 0.0 or p > 1.0:
            raise ValueError(f"dropout probability has to be between 0 and 1, but got {p}")

    def forward(self, x):
        if self.training:
            mask = torch.bernoulli(torch.full_like(x, self.p)).bool()
            return torch.where(mask, torch.relu(x), -torch.relu(-x))
        else:
            return torch.where(x >= 0, self.p * x, (1 - self.p) * x)
```

Listing 1: PyTorch implementation of the AID.

We provide the PyTorch implementation of the simplified version of AID in Listing 1, corresponding to Algorithm 2. Note that the negative ReLU function $\bar{r}(x)$ is implemented as $-r(-x)$, allowing for an efficient expression using standard ReLU operations. For all supervised learning experiments, we simply replaced all ReLU activations in the network with the AID module above. More extensive experiments are required, but intuitively, since AID can preserve negative preactivation values unlike ReLU, we recommend replacing maxpooling layers with average pooling layers in model, as discussed in Section F.3.

In the reinforcement learning experiments described in Section 5.2, an additional consideration is necessary. In general, when using dropout during policy rollouts, maintaining the model in training mode (i.e., with dropout active) can promote exploration by producing stochastic Q-values for the same state. However, in our setting, we found that this behavior introduced instability during training. Therefore, we set the model to evaluation mode when selecting greedy actions during rollouts, ensuring deterministic Q-value predictions. This strategy was applied consistently to both Dropout and AID. Investigating how such design choices affect the behavior of dropout-based methods in reinforcement learning remains an interesting direction for future work. All other aspects of the training procedure followed the standard DQN framework.

