# OpenReview forum: "Activation by Interval-wise Dropout: A Simple Way to Prevent Neural Networks from Plasticity Loss"
_ICML.cc/2025/Conference — ICML 2025 poster_

### Official Review · Reviewer_PypX · 2025-03-13

**Overall Recommendation:** 4

**Summary:**

This paper presents a new method for addressing loss of plasticity. The authors investigate why Dropout doesn’t help with loss of plasticity, although it helps with generalization. Through empirical work, the authors pinpoint the causes and introduce AID, an improvement over Dropout that maintains plasticity and improves generalization. The authors show their method's effectiveness in a wide range of experiments.

**Claims And Evidence:**

The paper is well-written and easy to follow. I like how the authors decided to show the problem first with Figure 1 and tried to convince the reader first before introducing the algorithm itself. The results show the effectiveness of AID on performance. The authors also complement their empirical work with some theoretical analysis of why AID is able to address loss of plasticity.

**Essential References Not Discussed:**

N/A

**Experimental Designs Or Analyses:**

The authors use standard benchmarking tasks that are widely used and accepted by the research community. I don’t see any issues in their experimental design or analysis.

**Methods And Evaluation Criteria:**

The authors considered standard tasks for plasticity like input/output permuted MNIST, continual CIFAR10/CIFAR100/TinyImageNet. The authors also included challenging settings like incremental class learning with CIFAR100 and TinyImageNet in addition to reinforcement learning with high-replay-ratio DQN.

**Other Comments Or Suggestions:**

- There is room to improve the introduction. Overfitting and loss of plasticity are often correlated, thus, most algorithms addressing overfitting would also address loss of plasticity. However, dropout (a method that addresses overfitting) does not address loss of plasticity. Thus, it’s interesting to understand why.

- The appendix refers to the method DIA instead of AID, which needs to be fixed.

- References with missing years:
  - (Shin et al.)
  - (Delfosse et al.).

**Other Strengths And Weaknesses:**

The paper is well-written and well-motivated. The performance gain when using AID seems to be significant and superior to other methods, even though it is a simple method. The approach is easy to implement, which is a plus. All in all, I would like to see this paper published. There are a few issues that can be addressed easily by the authors. If the authors made efforts towards fixing the issues, I’m going to raise my score to reflect the changes.

The main weaknesses I see in the paper:
- It’s not clear how AID is sensitive to its hyperparameter. The authors need to be transparent about this information.
- The comparison with dropout in the RL experiment is missing (Figure 6).

**Questions For Authors:**

> “During the testing phase of traditional dropout, each weight is scaled by the probability $1−p$, which is the chance that a node is not dropped. Similarly, in AID, as each interval has a unique dropout probability, the pre-activation values within each interval $I_j$ are scaled by $1−p_j$ for testing”

> “Dropout layer with probability p scales the value by $1/(1−p)$ at training phase, so it acts as an identity function during the test phase. However, since AID functions as a non-linear activation, we scale the value at test phase”

Shouldn’t the scaling be $1/(1-p)$ for dropout in the first paragraph and by $1/(1-p_j)$ for AID in the second paragraph?

Additionally, can the authors share why they are not scaling in Algorithm 2? Also, why the scaling is $1-p_j$ in Algorithm 1 instead of $1/(1-p_j)$
\
\
\
Another question: how would AID perform under the streaming supervised learning setting or the streaming reinforcement  learning setting?

**Relation To Broader Scientific Literature:**

The paper provides a comprehensive comparison of many methods addressing loss of plasticity. In addition, they related their method to similar ideas in the literature, namely DropReLU, CRELU, and RRELU. I think that answering the question of why Dropout doesn’t address loss of plasticity would interest a significant number of researchers.

**Theoretical Claims:**

I didn't check the correctness of the proofs of the theoretical claims.

---

> ### Author Rebuttal · Authors · 2025-04-01
>
> We thank the reviewer for their thoughtful comments and their positive assessment of the paper. We appreciate the recognition of our motivation, empirical scope, and clarity, and we respond below to the concerns raised.
>
> ---
> > It’s not clear how AID is sensitive to its hyperparameter. The authors need to be transparent about this information.
>
> We agree and will include a sensitivity analysis in the camera-ready version. Below, we report the mean and standard deviation of final accuracy for various AID hyperparameter settings (*p*) in the **continual full** setup, which we consider representative. We use the same seeding procedure as in the main paper. We report $p=0.6\sim 0.9$, as $p=0.5$ yields a linear network and $p=1$ gives ReLU. We exclude $p<0.5$ due to symmetry around 0.5.
> | *p* (Hyperparameter) | 0.6 | 0.7 | 0.8 | 0.9 | Best Baseline |
> | --- | --- | --- | --- | --- | --- |
> | CIFAR10 (CNN) | 0.601 ± 0.004 | 0.700 ± 0.004 | 0.746 ± 0.001 | **0.755 ± 0.004** | 0.717 ± 0.003 |
> | CIFAR100 (Resnet-18) | 0.5814 ± 0.002 | **0.640 ± 0.002** | 0.631 ± 0.003 | 0.600 ± 0.003 | 0.578 ± 0.005 |
> | TinyImageNet (VGG-16) | 0.438 ± 0.002 | **0.508 ± 0.0004** | 0.494 ± 0.004 | 0.454 ± 0.002 | 0.435 ± 0.001 |
>
> As shown, AID mostly outperforms the baselines across a range of hyperparameters, though some variance is expected. Additionally, as shown in Table 3, we used a very small hyperparameter search space for the **generalizability** and **RL** experiments—where AID still performs robustly. This suggests AID works well without extensive tuning.
>
> ---
> > The comparison with dropout in the RL experiment is missing (Figure 6).
>
> Dropout is not commonly used in deep RL due to its high variance and unstable training behavior. We also found that Dropout offered no consistent advantage over the vanilla model, which is why it was excluded from Figure 6. However, given that the paper discusses Dropout’s role in plasticity, we agree that including its RL results would be valuable.
>
> Below, we report the mean and standard deviation of final **human-normalized scores** for three games identified in Appendix G.3.2 as suffering from plasticity loss:
> | Game\Method | Vanilla | Dropout | AID |
> | --- | --- | --- | --- |
> | Asterix | 0.115 ± 0.007 | 0.180 ± 0.025 | **0.318 ± 0.054** |
> | BeamRider | 0.107 ± 0.021 | 0.124 ± 0.009 | **0.242 ± 0.052** |
> | DemonAttack | 0.122 ± 0.039 | 0.155 ± 0.052 | **0.714 ± 0.245** |
>
> While Dropout shows minor gains over vanilla in some cases, the improvements are negligible and likely reflect ensembling effects rather than plasticity preservation. In contrast, AID demonstrates consistent and substantial improvements, reinforcing its plasticity-related benefits.
>
> ---
> > There is room to improve the introduction…
>
> We believe this refers to the paragraph beginning *“Widely used methods…”* on page 2. We agree that including the relationship between the overfitting and plasticity loss can be made clearer, and including data augmentation as an example—as the reviewer suggests—would enhance the explanation. We will revise this in the camera-ready version.
>
> ---
> > The appendix refers to the method DIA instead of AID, which needs to be fixed.
> >
>
> > References with missing years [...]
>
> Thank you for pointing these out. We will correct these issues in the final version.
>
> ---
> > “Shouldn’t the scaling be 1/(1−p) for Dropout...?”
>
> We appreciate the reviewer’s attention to detail here. The confusion likely stems from different ways in implementing Dropout. To clarify:
>
> - **Standard implementation** [1]: no scaling at train time; multiply by $1 - p$ at test time.
> - **Inverted implementation** (e.g., PyTorch’s `nn.Dropout`): scale by $1/(1 - p)$ at train time; no scaling at test time.
>
> In our case, applying **inverted Dropout** to AID would make it act as an identity function during inference, removing non-linearity. Thus, we adopt **standard Dropout** way, which allows AID to maintain its activation function properties at test time via a modified leaky ReLU $r_p$.
>
> Algorithm 2 may appear not to scale, but the scaling is implicitly applied via $r_p$ during inference. We hope this helps in understanding how AID works.
>
> ---
> > How would AID perform under streaming supervised or RL settings?
>
> We thank the reviewer for the interesting question. Though we haven’t explicitly tested AID in streaming, our results show that AID effectively preserves plasticity in several challenging settings. This suggests it may also be helpful in streaming contexts.
>
> However, streaming learning often involves **catastrophic forgetting** in addition to plasticity loss. Since our work does not focus on forgetting mitigation, the effectiveness of AID in streaming scenarios would likely depend on the relative impact of forgetting versus plasticity loss in those settings. We believe this is a promising direction for future investigation.
>
> ---
> [1] Srivastava et al., Dropout: a simple way to prevent neural networks from overfitting, JMLR 2014

---

> > ### Comment · Reviewer_PypX · 2025-04-07
> >
> > I would like to thank the authors for their response. My concerns are addressed and therefore I increased my score accordingly.

---

### Official Review · Reviewer_KPJu · 2025-03-14

**Overall Recommendation:** 4

**Summary:**

The paper proposes AID, a novel activation function that generalizes Dropout to intervals, where it can be applied with different probabilities to different intervals of the activations. In its simplified version, it has 2 intervals (positive and negative) and can be interpreted as an interpolation between ReLU and a linear activation.
It claims that
1) AID effectively tackles plasticity loss in continual learning and reinforcement learning, unlike Dropout.
2) This is achieved by a regularization towards a linear regime.
3) AID also improves generalization in supervised learning.

## update after rebuttal
All my concerns have been addressed. It also seems that the other reviewers' concerns have been addressed (apart from apparently Reviewer  AVaP, with whom we can engage in discussions if they feel otherwise). I maintain my score from the end of the rebuttal.

**Claims And Evidence:**

The claims are supported by evidence, which I do not find fully sufficient.

1.a) continual learning - The claim is supported by Figure 2 showing the benefits of AID on plasticity loss and generalizability, unlike dropout.

1.b) Reinforcement learning - Figure 6 does not show benefits on plasticity in my understanding. I would expect to see the vanilla method collapse but AID not to isolate plasticity as a cause. Otherwise, it's not clear that the benefits of AID come from better plasticity.

2.) This is addressed in Theorem 4.1. The notation is not clear, and the interpretations of the results do not help in supporting the claim.

3.) The claim is supported by Table 1.

**Essential References Not Discussed:**

The paper references the important literature combining dropout and ReLU and other attempts at mitigating plasticity loss.

**Experimental Designs Or Analyses:**

Figure 4. Vanilla does not seem to collapse; how are the benefits in plasticity isolated from the benefits in generalization?

**Methods And Evaluation Criteria:**

The methods and evaluations in the paper are very well chosen and performed.
The paper categorizes and compares AID to multiple SoTA methods proposed to mitigate plasticity loss.
It uses supervised learning, continual learning, and reinforcement learning experiments.
All the experiments have enough statistical significance.

**Other Comments Or Suggestions:**

No other comments.

**Other Strengths And Weaknesses:**

+The explanation of why dropout does not prevent plasticity is beneficial.
-Only performance is used to diagnose plasticity loss. The paper would be stronger by incorporating other metrics like feature rank, weight norm, activation norms, etc.

**Questions For Authors:**

No other questions.

**Relation To Broader Scientific Literature:**

The paper provides a good discussion of the broader scientific literature relating to plasticity and activation functions.

**Theoretical Claims:**

I did not check the proofs of the theoretical claims which are in the Appendix.

---

> ### Author Rebuttal · Authors · 2025-04-01
>
> We thank the reviewer for their detailed feedback and thoughtful questions. We appreciate your recognition of the motivation, experimental setup, and analysis, and we address your concerns below.
>
> ---
>
> > 2. b) Reinforcement learning [...] I would expect to see the vanilla method collapse [...]
> >
>
> Reinforcement learning settings do not induce non-stationarity as severely as random-label or permuted MNIST. Indeed, prior studies with similar setups [1,2] also do not report full IQM score collapse. Instead, they demonstrate the presence of plasticity loss in reinforcement learning by showing that addressing it—through improvements in various indicators of plasticity such as dormant neurons and effective rank—leads to enhanced performance. Our experiments use the same setting—four times higher replay ratio (RR) than in typical DQN—which has been shown to accelerate plasticity loss [1,2,3], and is widely used in recent plasticity research [1,2,4].
>
> Game-specific results provide additional insight: AID outperforms the baseline in **DemonAttack**, **SpaceInvaders**, and **Asterix**, where the vanilla model struggles—suggesting plasticity loss. In contrast, for games like **Pong** and **Qbert**, where the plasticity loss is not the main bottleneck for the performance , AID does not benefit from mitigating plasticity loss. While other factors may contribute to performance,  the results of the trainability and high RR experiments support our interpretation that AID mitigates plasticity loss.
>
> ---
>
> > 3. “Theorem 4.1. The notation is not clear, and the interpretations of the results do not help in supporting the claim.”
> >
>
> We appreciate the reviewer’s feedback. We acknowledge that the current notation and interpretation in Theorem 4.1 could be clearer, and we will revise these in the camera-ready version.
>
> Theorem 4.1 shows that the lower bound of the loss function under AID, denoted as L_{\text{AID}*p}, can be decomposed into the loss under a modified leaky ReLU, L*{r_p}, and a regularization term  that encourages **linearity** in the network. As the hyperparameter p approaches 0.5, AID imposes stronger regularization toward linear behavior.
>
> Prior work [5] has shown that **linear networks do not suffer from plasticity loss**, suggesting that the linearity-inducing effect of AID (as formalized in Theorem 4.1) directly contributes to mitigating plasticity loss.  In contrast, naive dropouts cannot induce the regularization term associated with linearity (Appendix C.2), which may explain its ineffectiveness at preserving plasticity.
>
> ---
>
> > Figure 4. Vanilla does not seem to collapse; how are the benefits in plasticity isolated from the benefits in generalization?
> >
>
> Accuracy collapse is the indicator of trainability. The experiment in Figure 4 focuses on **generalizability**, evaluated by comparing the performance of the vanilla model to that of a fully reset model. A performance gap here indicates **loss of generalizability**, a form of plasticity degradation.
>
> A method with poor generalizability but good generalization may initially perform well, but lose its advantage over time. In contrast, AID maintains consistent performance throughout training. As shown in Section 3.2, Dropout enhances generalization performance but does **not** improve generalizability. AID, by narrowing the performance gap, demonstrates its ability to preserve generalizability.
>
> ---
>
> > - Only performance is used to diagnose plasticity loss. […]
> >
>
> In Appendix G.1.2, we evaluate multiple **plasticity metrics**, including **dormant neuron ratio** [1], **average unit sign entropy** [5], and **effective rank** [2], comparing vanilla, Dropout, and AID. In Appendix G.1.3, we also analyze **pre-activation distribution**, which is one of the causes that induces plasticity loss [6].
>
> These analyses show that:
>
> - Dropout does not improve plasticity indicators relative to the vanilla baseline.
> - AID shows favorable trends across all metrics.
> - AID causes significantly less change to pre-activation distributions than Dropout.
>
> Thus, our claims are supported by a combination of **performance metrics** and **plasticity metrics**.
>
> ---
>
> [1] Sokar et al., *The Dormant Neuron Phenomenon in Deep Reinforcement Learning*, ICML 2023
>
> [2] Kumar et al., *Implicit Under-Parameterization Inhibits Data-Efficient Deep RL*, ICLR 2021
>
> [3] Nikishin et al., *The Primacy Bias in Deep Reinforcement Learning*, ICML 2022
>
> [4] Elsayed et al., *Weight Clipping for Deep Continual and Reinforcement Learning*, RLC 2024
>
> [5] Lewandowski et al., *Plastic Learning with Deep Fourier Features*, ICLR 2025
>
> [6] Lyle et al., *Disentangling the Causes of Plasticity Loss in Neural Networks*, arXiv 2024

---

> > ### Comment · Reviewer_KPJu · 2025-04-05
> >
> > The authors addressed all my concerns. Raising my score from 2 to 4.

---

### Official Review · Reviewer_AVaP · 2025-03-16

**Overall Recommendation:** 2

**Summary:**

The proposed method in this paper is motivated by the characteristic of dropout, poor trainability and good generalizability. By enhancing the trainability of dropout in AID, the model can have both good trainability and generalizability. The main point of AID is separating the neurons into positive and negative part, and using different masks to utilize both positive and negative neurons. As a result, the network can obtain better trainability than using naive dropout. In the experiment, AID outperforms most of the baselines in warm-start scenario, and is also effective on reinforcement learning.

**Claims And Evidence:**

The main claims are not quite persuasive. The motivation experiments are not related to the results in the experiment section.

**Essential References Not Discussed:**

N/A

**Experimental Designs Or Analyses:**

Experiment designs are valid.

**Methods And Evaluation Criteria:**

The evaluation criteria does make sense.

**Other Comments Or Suggestions:**

Already mentioned in the above section.

**Other Strengths And Weaknesses:**

Strengths

1. In the warm-start scenario experiment, AID achieves much higher accuracy than other baselines. Furthermore, AID is also effective on reinforcement learning


Weaknesses

1. As mentioned in this paper, the main disadvantage of using dropout is its low trainability in non-stationary target scenario, and the authors show the degradation of the trainability in permuted MNIST and random label MNIST experiment. However, in the experiment section, the authors only carried out the experiment on warm-start scenario, and it is hard to figure out whether the trainability issues are really occur in this scenario. If it is not, does the performance increase are from enhanced trainability? or just from the generalization capability of AID? Since the initial test accuracy of AID is much higher than naive dropout, it is hard to know that AID can actually resolve the trainability issue.

2. It may not be a fair comparison in warm-start scenario between AID and other baselines since the initial test accuracy of AID is much higher than other baselines. To make the comparison be fair, all the methods have same generalizability at the learning phase, and only additional treatment are applied when the number of training data increases.

3. If AID is applied only when the experiment transition occurs similar to DASH [1] (i.e. apply AID when the number of training data increases), does AID still achieve better performance than others? If it does not increase the performance, we can think that applying AID is just giving generalization capability similar to other generalization techniques (e.g. data augmentation, or batch normalization) which are always effective regardless of the learning scenario such as warm-start or non-stationary target experiments.

4. As in theorem 4.1, the authors said applying AID can inject the linearity which can mitigate the plasticity loss (i.e. trainability). However, I wonder this theorem is actually aligned with the experiment results in warm-start scenario.

[1] Shin et. al., DASH: Warm-Starting Neural Network Training in Stationary Settings without Loss of Plasticity, NeurIPS, 2024

**Questions For Authors:**

Already mentioned in the Strengths And Weaknesses section.

**Relation To Broader Scientific Literature:**

N/A

**Theoretical Claims:**

The theoretical results are not aligned to the empirical results.

---

> ### Author Rebuttal · Authors · 2025-04-01
>
> We sincerely thank the reviewer for the thoughtful and constructive feedback. First, we clarify a potential misconception: **Dropout does not significantly improve generalizability.**
>
> > “The proposed method in this paper is motivated by the characteristic of dropout, poor trainability and **good generalizability**.”
> >
>
> In the literature, generalization performance in deep learning and generalizability [1] in plasticity have different meanings. This is addressed in Section 3.2 and illustrated in Figure 2 (right). While Dropout slightly improves performance, it does not reduce the performance gap between cold-start and warm-start, which signals loss of generalizability. In contrast, AID narrows this gap, indicating better generalizability. Thus, we show that Dropout is limited in both trainability and generalizability, whereas AID improves both. We appreciate the opportunity to clarify this important distinction and will revise the wording in the camera-ready version to avoid confusion.
>
> ---
>
> We address the reviewer’s concerns below.
>
> > 1. [...] authors only carried out the experiment on warm-start scenario [...] it is hard to figure out whether the trainability issues are really occur in this scenario. [...]
> >
>
> While the reviewer notes that our experiments focus on the warm-start scenario, our paper evaluates both **trainability** and **generalizability**, as shown in Section 5.1.1 and Section 5.1.2, respectively. Additional trainability results are included in Appendix G.1. Importantly, trainability issues typically arise in **non-stationary settings** (e.g., permuted or random-label MNIST) [5], while generalizability issues occur in **stationary continual settings** such as warm-start and class-incremental learning [2,3]. Prior work [1,4,5] has established these tasks as **standard benchmarks** for studying each form of plasticity loss, and we follow the same practice in our experiments.
>
> > 1. [...] initial test accuracy of AID is much higher than naive Dropout [...] To make the comparison be fair, all the methods have same generalizability at the learning phase [...]
> >
>
> We understand the reviewer’s concern about fairness in comparison. The initial accuracy difference stems from hyperparameter choice. All methods were tuned based on final accuracy, reflecting typical practice. For instance, we found the Dropout setting with similar initial accuracy to AID, but their final accuracy gets lower (e.g., continual full, ResNet-18, table below). This highlights AID’s strength in consistently retaining generalization throughout training.
>
> | Epoch | 100th | 1000th |
> | --- | --- | --- |
> | Dropout p=0.1 | 0.255 ± 0.005 | 0.5395 ± 0.003 |
> | Dropout p=0.3 | 0.136 ± 0.023 | 0.5398 ± 0.005 |
> | AID p=0.7 | 0.289 ± 0.005 | 0.638 ± 0.001 |
>
> > 2.  [...] and only **additional treatment are applied when the number of training data increases.**
>
> This is a valuable point. We distinguish two classes of methods for generalizability in warm-start:
>
> - Applied only as data comes in: S&P [3], head reset, DASH [2]
> - Apply continuously: H&T [1], Fourier activation [4], AID
>
> AID belongs to the second category. These methods are advantageous as they do not require task boundary information [6], making them more practical—especially in settings like RL where task boundaries are implicit or unknown. They can also complement transition-based ones, making AID broadly applicable.
>
> > 3. If AID is applied only when the experiment transition occurs [...] does AID still achieve better performance than others? [...]
>
> Since AID is implemented as an activation function, it is structurally unsuitable for being applied only at task transitions.
>
> > 4. I wonder this theorem is actually aligned with the experiment results in warm-start scenario.
>
> We agree this is a subtle point. Theorem 4.1 supports AID’s ability to improve trainability by regularizing toward linear behavior. It does not directly explain the generalizability improvements observed in warm-start settings.
>
> However, prior work [4] has empirically shown that encouraging linearity can also lead to improved generalizability in several continual learning scenarios. Similarly, AID’s ability to inject linearity appears to yield benefits not only for trainability but also for generalizability in our experiments. This contrasts with other methods for trainability, which often fail to improve generalizability [1]—further supporting this interpretation.
>
> ---
>
> [1] Lee et al., Slow and Steady Wins the Race: Maintaining Plasticity with Hare and Tortoise Networks, ICML 2024
>
> [2] Shin et al., DASH: Warm-Starting Neural Network Training in Stationary Settings without Loss of Plasticity, NeurIPS 2024
>
> [3] Ash et al., On Warm-Starting Neural Network Training, NeurIPS 2020
>
> [4] Lewandowski et al., Plastic Learning with Deep Fourier Features, ICLR 2025
>
> [5] Lyle et al., Disentangling the Causes of Plasticity Loss in Neural Networks, arXiv 2024
>
> [6] Aljundi et al., Task-Free Continual Learning, CVPR 2019

---

> > ### Comment · Reviewer_AVaP · 2025-04-06
> >
> > Thank you for the effort on the authors' rebuttal comment.
> >
> > After carefully reading the comments, I still wonder comparing AID with other baselines is still fair.
> >
> > Since applying AID can boost the performance on single task, it is hard to figure out the excessive test accuracy comes from resolving the over-fitting problem in warm-starting scenario or just improving a single task performance using a proper regularization technique.
> >
> > Though the authors give additional experiments, as we can see in the bottom figure in Figure 4, only using dropout outperforms other baselines, and I don't think this resuls shows the dropout can resolve the over-fitting problem in warm-start scenario.
> >
> > Based on the rebuttal comments, I will keep my score.

---

> > > ### Author Response · Authors · 2025-04-07
> > >
> > > We appreciate the reviewer’s continued engagement and thoughtful remarks following our rebuttal.
> > >
> > > We understand the reviewer’s concern that AID may appear to alleviate plasticity loss (PL) in the warm-start scenario simply due to its improved generalization performance. First, we would like to clarify that there is strong evidence in Figure 2 (right) to support the claim that the improved performance of AID comes from addressing generalizability loss, not generalization improvement. The obvious way to determine if a model has generalizability loss is to compare its performance to a model with all parameters reset when new data is added, since a freshly initialized model has perfect plasticity. Using this way, previous studies for plasticity loss [1,2,3] have shown that plasticity loss exists in their baseline models and their proposed model solves the plasticity loss.
> > >
> > > Based on this approach, the greater the difference between the performance of a model with AID and a full reset (resetting all parameters whenever new data comes in, also termed as cold start) and the performance of a model with AID alone, the greater the loss of generalizability.
> > >
> > > Figure 2 (right) shows that the accuracy gap between AID+full reset (cold start) and AID is very low (3.3%p), and the accuracy gap between Dropout+full reset (cold start) and Dropout is high (10.1 percentage points). In particular, the accuracy gap of Dropout is similar to that of the vanilla model (10.9%p), indicating that Dropout does not address the generalizability loss at all. On the other hand, AID has an accuracy gap of 3.3%p, which is very low compared to the vanilla model (10.9%p) and Dropout (10.1%p). These results show that AID's performance improvement comes from addressing generalizability loss, while Dropout does not.
> > >
> > > To provide further evidence, we extended the experiment in Figure 2 (right), where data is added only once, to compare performance differences in the continual full setting, where new data is added multiple times. We analyzed the difference in final accuracy for each chunk, computed as the performance gap between models trained with and without full reset. We adopted three methods—Vanilla, Dropout, and AID—in the **continual full** setting. A positive difference indicates that full reset improves performance, suggesting the presence of generalizability loss. For each method, hyperparameters were independently tuned to maximize final performance both with and without reset. The number of seeds follows our original setup, and we omit standard deviations here for readability.
> > >
> > > ### CIFAR10 (CNN)
> > >
> > > | Acc (%p) \ Chunk | 1 | 2 | 3 | 4 | 5 | 6 | 7 | 8 | 9 | 10 |
> > > | --- | --- | --- | --- | --- | --- | --- | --- | --- | --- | --- |
> > > | (Vanilla+Reset) - Vanilla | 0.11 | -0.31 | 0.76 | 1.39 | 2.25 | 3.06 | 3.22 | 3.68 | 4.23 | 4.96 |
> > > | (Dropout+Reset) - Dropout | 2.49 | 2.42 | 1.40 | 1.55 | 1.04 | 1.07 | 1.22 | 1.48 | 1.17 | 1.10 |
> > > | (AID+Reset) - AID | 1.47 | -0.39 | -0.67 | -0.45 | -0.43 | -1.37 | -0.93 | -0.74 | -0.77 | -1.32 |
> > >
> > > ### CIFAR100 (Resnet-18)
> > >
> > > | Acc (%p) \ Chunk | 1 | 2 | 3 | 4 | 5 | 6 | 7 | 8 | 9 | 10 |
> > > | --- | --- | --- | --- | --- | --- | --- | --- | --- | --- | --- |
> > > | (Vanilla+Reset) - Vanilla | 0.57 | 4.42 | 7.58 | 8.93 | 8.56 | 10.29 | 12.60 | 11.69 | 11.61 | 12.08 |
> > > | (Dropout+Reset) - Dropout | 10.80 | 4.98 | 4.69 | 5.83 | 6.37 | 6.82 | 7.66 | 7.65 | 7.56 | 8.29 |
> > > | (AID+Reset) - AID | -1.36 | 0.48 | 0.97 | 1.91 | 1.97 | 1.83 | 2.19 | 2.15 | 1.85 | 2.43 |
> > >
> > > ### TinyImageNet (VGG-16)
> > >
> > > | Acc (%p) \ Chunk | 1 | 2 | 3 | 4 | 5 | 6 | 7 | 8 | 9 | 10 |
> > > | --- | --- | --- | --- | --- | --- | --- | --- | --- | --- | --- |
> > > | (Vanilla+Reset) - Vanilla | -0.27 | 1.54 | 5.01 | 4.98 | 9.06 | 10.12 | 11.60 | 13.65 | 13.5 | 14.65 |
> > > | (Dropout+Reset) - Dropout | -5.22 | 1.37 | 1.86 | 5.17 | 6.51 | 8.48 | 3.00 | 8.62 | 9.71 | 9.17 |
> > > | (AID+Reset) - AID | -0.07 | -1.53 | -1.30 | -2.2 | -1.57 | -1.46 | -1.59 | -1.37 | -0.98 | -0.43 |
> > >
> > > From these results, we observe that while Dropout provides modest gains in generalizability over Vanilla in smaller models, it fails to close the gap with full reset in larger models. In contrast, AID outperforms (AID+full reset) in both CNN and VGG-16, and substantially narrows the gap on ResNet-18. This indicates that AID not only enhances generalization capacity (as discussed in Section 5.3), but also preserves generalizability, which is crucial in warm-start and other stationary continual learning settings.
> > >
> > > We hope these additional results clarify the issue that AID’s improved performance in warm-start scenarios results solely from enhanced generalization capacity.
> > >
> > > ---
> > >
> > > [1] Ash et al., On Warm-Starting Neural Network Training, NeurIPS 2020
> > >
> > > [2] Nikishin et al., The Primacy Bias in Deep Reinforcement Learning, ICML 2022
> > >
> > > [3] Nikishin et al., Deep Reinforcement Learning with Plasticity Injection, NeurIPS 2023

---

### Decision · Program_Chairs · 2025-05-01

**Decision:**

Accept (poster)

**Comment:**

This paper introduces a variation of dropout that leads to modest improvements in trainability and generalizability for neural networks.
Overall, there was minimal but positive enthusiasm for the work.